# Schema representations in distinct brain networks support narrative memory during encoding and retrieval

Rolando Masís-Obando[1]*, Kenneth A Norman[1,2], Christopher Baldassano[3]

[1]Princeton Neuroscience Institute, Princeton, United States; [2]Department of Psychology, Princeton University, Princeton, United States; [3]Department of Psychology, Columbia University, New York, United States

**Abstract** Schematic prior knowledge can scaffold the construction of event memories during perception and also provide structured cues to guide memory search during retrieval. We measured the activation of story-specific and schematic representations using fMRI while participants were presented with 16 stories and then recalled each of the narratives, and related these activations to memory for specific story details. We predicted that schema representations in medial prefrontal cortex (mPFC) would be correlated with successful recall of story details. In keeping with this prediction, an anterior mPFC region showed a significant correlation between activation of schema representations at encoding and subsequent behavioral recall performance; however, this mPFC region was not implicated in schema representation during retrieval. More generally, our analyses revealed largely distinct brain networks at encoding and retrieval in which schema activation was related to successful recall. These results provide new insight into when and where event knowledge can support narrative memory.

## Editor's evaluation

This paper reports a methodologically rigorous investigation into the neural mechanisms supporting encoding and retrieval of specific and general information in the context of memory schemas for events or 'scripts.' Its findings will be of general interest to neuroscientists and cognitive psychologists who work both with typical young adults (as studied in this paper) and in particular populations (e.g., development and/or aging; patients with brain damage). The work is particularly comprehensive in how it links both specific and general narrative representation at both encoding and retrieval with later memory behavior. This comprehensive treatment is a notable strength.

*For correspondence:
rmasis@princeton.edu

Competing interest: The authors declare that no competing interests exist.

## Introduction

How do we remember real-world events? Over the past half-century, the cognitive psychology literature has shown that we leverage event schemas – our knowledge of how events generally unfold – to support memory for specific details from those events (for reviews of early work, see *Graesser and Nakamura, 1982*; *Alba and Hasher, 1983*; *Brewer and Nakamura, 1984*; for more recent cognitive neuroscience studies, see *van Kesteren et al., 2012*; *Ghosh and Gilboa, 2014*; *Schlichting and Preston, 2015*; *Gilboa and Marlatte, 2017*; *Preston and Eichenbaum, 2013*; *Wang and Morris, 2010*). For example, when we go into a restaurant, we can anticipate a stereotyped sequence of events that includes getting seated, ordering food, and eating (*Bower et al., 1979*). The cognitive psychology literature has demonstrated that knowledge of this 'restaurant script' can help memory in at least two possible ways: At encoding, the restaurant script can provide a scaffold onto which we can

**eLife digest** Our day-to-day experiences are incredibly complex, so how does the brain remember them? Cognitive scientists have shown that memories rely on knowledge of common events that we have experienced before. Think about going to a restaurant: you arrive, you find a table, you order food, and then you eat. This kind of predictable sequence is called a schema. When humans make memories, our brains use schemas like these as scaffolding. They take a basic pattern constructed from past experience and fill it in with the specific details of an event. When memories are recalled, our brains use schemas as step-by-step guides to remember the events in the right order.

Most research so far on how the brain uses schemas for memory has involved showing participants pictures or words and then testing their memory by asking 'true or false' questions. This revealed that a brain area called the medial prefrontal cortex plays an important role in creating and retrieving memories for items related to a schema. But, studies have not yet assessed exactly how the brain uses schemas to understand and remember a long, realistic event that unfolds over several minutes.

To answer this question, Masís-Obando et al. scanned people's brains while they watched or listened to clips of two familiar experiences: eating at a restaurant or catching a flight at an airport. Then, the participants were scanned while they tried to retell each story in their own words. The volunteers were graded based on how many details they recalled. The scans showed that when volunteers' medial prefrontal cortex kept track of the schema throughout the whole time that an event was happening, they were more likely to score well on the memory test. But it wasn't necessary for medial prefrontal cortex to hold the schema in mind when remembering the story. Instead, a different set of brain regions maintained schema information during successful remembering.

This study reveals new information about how memories and schemas work that could help explain why people develop problems making or recalling memories in diseases such as Alzheimer's. The findings could also be used to help people make experiences or stories more memorable.

attach specific event details (e.g., *Bransford and Johnson, 1972*; *Alba and Hasher, 1983*; *Abbott et al., 1985*; *Tompary and Thompson-Schill, 2021*; *McClelland et al., 2020*); later, at retrieval, the restaurant script provides a structured way of cueing memory, by stepping through the various stages of the script in sequence (e.g., *Schank and Abelson, 1975*; *Anderson and Pichert, 1978*; *Bower et al., 1979*; *Alba and Hasher, 1983*; *Mandler, 2014*).

The goal of this study is to understand the neural mechanisms of how event schemas support memory for real-world, temporally extended events, both at encoding and at retrieval. To meet this objective, we track schema representations in the brain during both encoding and retrieval of temporally extended events, and then relate these neural measures to behavioral recall on a story-by-story basis. While there has been an explosion of recent neuroscientific research into how schemas benefit memory (*Maguire et al., 1999*; *van Kesteren et al., 2013*; *van Kesteren et al., 2010a*; *van Kesteren et al., 2020*; *van Kesteren et al., 2018*; *Spalding et al., 2015*; *Liu et al., 2018*; *Brod et al., 2015*; *Brod and Shing, 2018*; *van Buuren et al., 2014*; *Wagner et al., 2015*; *Bein et al., 2014*; *Schlichting and Preston, 2016*; *Tse et al., 2007*; *Tse et al., 2011*; *Webb et al., 2016*; *Gilboa and Marlatte, 2017*; *Raykov et al., 2021*; *Reagh et al., 2021*), most of this research has relied on univariate contrasts of brain activations evoked by schema-consistent vs. schema-inconsistent learning materials, rather than trying to track the degree to which schematic information is represented for individual stimuli. Also, existing studies have mostly looked at relatively simple forms of schematic knowledge (e.g., seashells at the beach vs. lamps at a playground; *McAndrews et al., 2016*) rather than knowledge about the structure of real-world, temporally extended events. Lastly, because existing paradigms have mostly tested memory with recognition or short associative recall tasks, the neural mechanisms of how schemas are instantiated and maintained during unconstrained memory search for naturalistic events have not been thoroughly explored; our use of free recall allowed us to address this gap in the literature.

The present study builds on our prior work (*Baldassano et al., 2018*), in which participants were scanned as they watched movies or listened to audio-adaptations of preexisting films, half of which followed a restaurant script and half of which followed an airport script. A key benefit of this paradigm is that it allowed us to identify sequences of neural patterns that are unique to particular stories (e.g.,

sequences of patterns that are reliably invoked by a particular airport narrative, more so than by other airport narratives) and sequences of patterns that represent the underlying script (e.g., sequences of patterns that are shared across different airport narratives, more so than across restaurant and airport narratives). *Baldassano et al., 2018* leveraged this to identify a range of areas that represented schematic information (i.e., restaurant vs. airport) in a modality-independent fashion. Of the regions of interest (ROIs) investigated, medial prefrontal cortex (mPFC) was the only one that was sensitive to the specific temporal order of events in a schema. Here, we extend the *Baldassano et al., 2018* results by analyzing neural and behavioral data from a separate phase of the experiment (not reported in the 2018 study) in which participants were scanned while freely recalling each of the 16 narratives. This allowed us to look at how schemas are represented in the brain during recall, and how neural measures of schema representation at encoding and recall are related to recall of specific story details, on a story-by-story basis.

Because mPFC has been frequently implicated in previous schema research (e.g., *van Kesteren et al., 2013*; *van Kesteren et al., 2010a*; *van Kesteren et al., 2020*; *van Kesteren et al., 2014*; *Baldassano et al., 2018*; *Raykov et al., 2020*; *Raykov et al., 2021*; *Reagh et al., 2021*) – in particular, with regard to integrating new knowledge into existing schemas (*Preston and Eichenbaum, 2013*; *Schlichting and Preston, 2015*; *Gilboa and Marlatte, 2017*; *Tse et al., 2007*; *Wang and Morris, 2010*; *van Kesteren et al., 2012*) – we predicted that robust mPFC schema representations at encoding would lead to improved subsequent memory for the narrative. Based on prior work implicating the hippocampus in schema representation (*van Kesteren et al., 2013*; *van Kesteren et al., 2020*; *van Kesteren et al., 2014*; *Brod et al., 2015*; *Liu et al., 2017*; *Raykov et al., 2020*; *Webb et al., 2016*; *van der Linden et al., 2017*; *Bonasia et al., 2018*), we also hypothesized that hippocampal schema representations at encoding would support subsequent memory; more specifically, based on work showing that hippocampus has a coarse-to-fine gradient of representations along its long axis (*Collin et al., 2015*; *Guo and Yang, 2020*; *Audrain and McAndrews, 2020*; *Poppenk et al., 2013*; *Brunec et al., 2018*; *Schlichting et al., 2015*; *Sekeres et al., 2018*), we predicted that anterior hippocampus (which has coarser and thus more general representations than posterior hippocampus) would contain schematic representations that contribute to subsequent memory, whereas posterior hippocampus would contribute to subsequent memory by representing story-specific details.

As described below, our prediction about mPFC was upheld: An anterior region of mPFC was among the network of cortical regions – also including left visual cortex, right lateral superior frontal gyrus (SFG), prostriata, and entorhinal cortex – where the degree of schema representation at encoding predicted subsequent memory for story details. Our prediction about hippocampus received partial support: While the degree of schema representation in anterior hippocampus during encoding showed a nonsignificant, positive numerical relationship to subsequent memory for story details, posterior hippocampus showed a negative correlation between schema representation at encoding and subsequent memory, and a positive correlation between the representation of story-specific details at encoding and subsequent memory – both of which are consistent with a role for posterior hippocampus in encoding story-specific (i.e., non-schematic) information. Interestingly, the set of regions where schema representation at encoding predicted recall of story details was mostly distinct from the set of regions where schema representation at retrieval predicted recall of story details – the latter analysis revealed a distinct network including bilateral visual cortex, right superior parietal lobule (SPL), bilateral middle frontal gyrus (MFG), bilateral medial SFG, bilateral parahippocampal cortex (PHC), left fusiform gyrus, right angular gyrus (AG), as well as bilateral posterior superior temporal sulcus, but notably *excluding* mPFC. This pattern of results provides converging neural support for the idea that schemas play different roles at encoding and retrieval in supporting memory for story details.

## Results

Our primary goal was to understand how we use schemas at encoding and recall to support memory for recently encoded naturalistic stories. To do this, we used 16 narratives that conformed to one of two schematic scripts (*Bower et al., 1979*): eating at a restaurant or catching a flight at an airport (*Figure 1*). Each narrative followed a four-event temporal structure specific to its schema (*restaurant stories*: entering the restaurant, being seated, ordering and eating food; *airport stories*: entering the airport, going through security, boarding at gate, and getting seated on plane). During the encoding

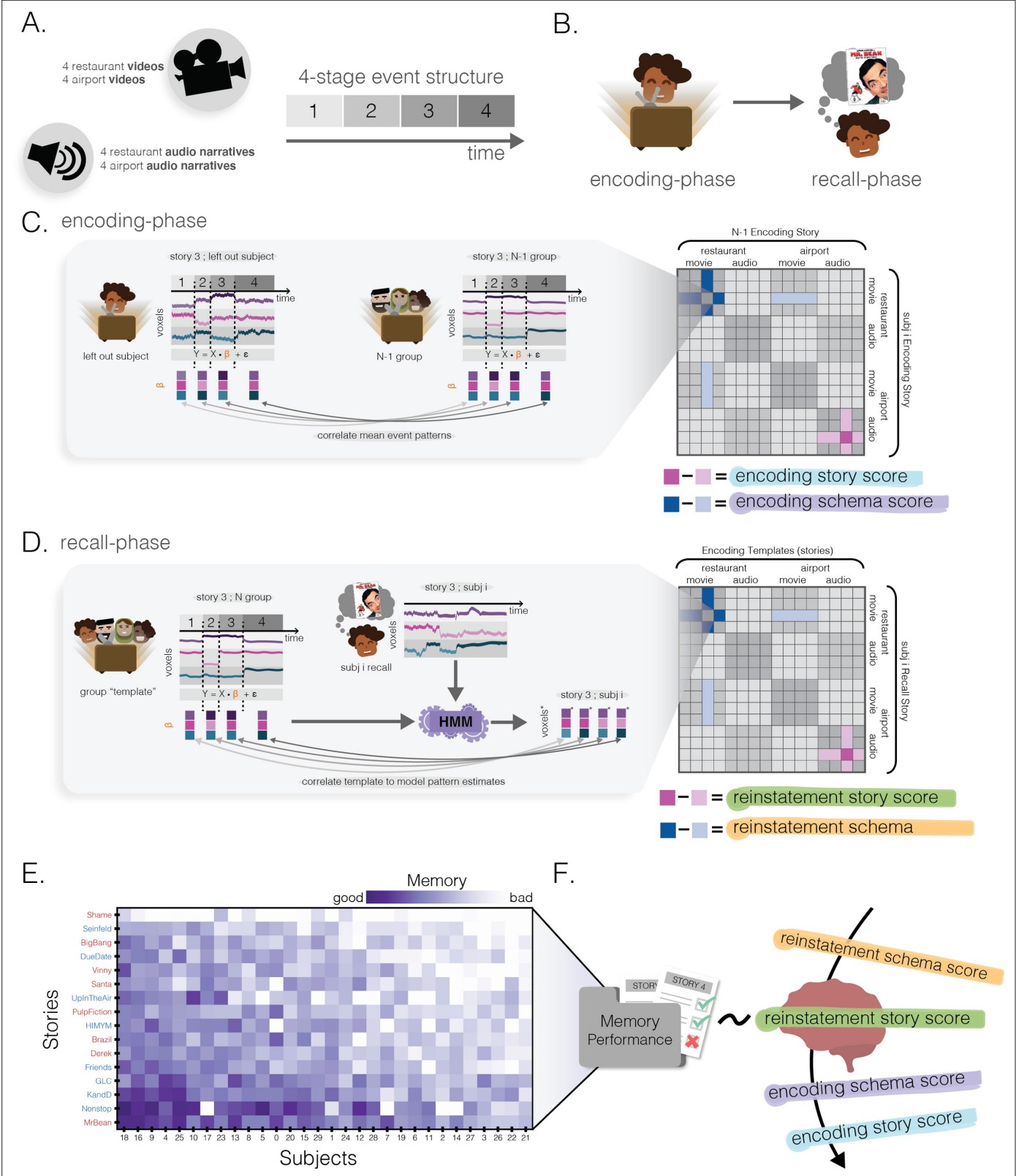

**Figure 1.** Methods. (**A**) Stimuli. There were a total of 16 narratives (audiovisual clips or spoken narration): eight restaurant narratives and eight airport narratives. Each narrative followed a four-event temporal structure specific to its schema (see text). (**B**) Experimental protocol. After participants encoded each of the narratives, they were then asked to freely recall each of them with a title cue only. (**C**) Encoding story and schema score. For each story in each participant, a spatial activity pattern was extracted for each of the four events in that story. We then computed, for each participant, the

*Figure 1 continued on next page*

*Figure 1 continued*

16 × 16 neural similarity matrix correlating the neural representations of each of the 16 stories in that participant and the neural representations of each of the 16 stories, averaged across the other participants (see text for details). For each story in each participant, we computed an *encoding story score* contrasting across-subject neural similarity to the same story (dark pink) vs. different stories from the same schema (light pink); we also computed an *encoding schema score* contrasting across-subject neural similarity to different stories from the same schema (dark blue) vs. different stories from the other schema (light blue). (**D**) Reinstatement story and schema score. We used hidden Markov models (HMMs) to measure the degree to which each of the 16 stories from the encoding phase was neurally reinstated during recall of a given story (see text for details). This process yielded a 16 story × 16 story neural reinstatement matrix for each participant. Analogously to (**C**), for each recall we computed a *reinstatement story score* (contrasting how well the same story's encoding pattern was reinstated vs. other stories from the same schema) and a *reinstatement schema score* (contrasting how well other stories from the same schema were reinstated vs. other stories from the other schema). (**E**) Behavioral memory performance. Every participant's free recall was scored using a rubric to measure the number of story-specific details the participant provided. This matrix has been sorted such that the most accurate recalls are in the bottom-left. Red and blue story labels indicate restaurant vs. airport narratives, respectively. (**F**) Predicting behavioral memory performance. We used the 4 scores derived from (**C**) and (**D**) (encoding story/schema and reinstatement story/schema) in four separate regression models to predict behavioral memory performance in (**E**).

phase, participants were scanned while they watched or listened to each of these 3 minute narratives. Afterwards, during the recall phase, participants were cued with the titles of each of the stories and were asked to freely verbally recall one story at a time.

## Neural story and schema scores

### Encoding scores

We derived two types of neural scores that reflected the extent to which story-specific and general schematic information were represented during encoding (*Figure 1C*). These scores were computed in both searchlights and specific ROIs (cortical ROIs: mPFC, posterior medial cortex (PMC), AG, PHC, and SFG; hippocampal ROIs: full hippocampus, anterior hippocampus, and posterior hippocampus). Within each story, we computed the mean spatial pattern evoked during each of the four events for each participant. Then, for each pair of stories (call them story A and story B), we applied leave-one-participant-out spatial intersubject correlation, correlating the four story A event patterns from the left-out participant with the four story B event patterns from the other participants. As in *Baldassano et al., 2018*, this correlation was computed in an event-wise fashion (correlating event 1 in story A with event 1 in story B, event 2 in story A with event 2 in story B, etc.), and then the four event-wise correlations were averaged together to obtain a single correlation score for the pair of stories. To measure the degree of story-specific representation at encoding for a participant experiencing a particular story, we computed an *encoding story score*, operationalized as the across-participant similarity to the representation of the *same* story, minus the average across-participant similarity to other stories from the same schema. To measure the degree of schematic representation at encoding, we computed an *encoding schema score*, operationalized as the average across-participant similarity to other stories from the same schema, minus the average across-participant similarity to other stories from the other schema. For all analyses reported below on our specific a priori ROIs, we report multiple comparisons Bonferroni-corrected p-values, such that p-values for cortical ROIs (n = 5) and hippocampal ROIs (n = 3) were scaled by 5 or 3, respectively, to uphold a significance level of alpha = 0.05.

Results from this encoding analysis were previously reported in *Baldassano et al., 2018* using a similar analysis pipeline. Encoding story scores were high across all of cortex (*Figure 2A*; q < 0.05), including all of our cortical and hippocampal ROIs (all p<0.01), with the strongest effects in posterior sensory regions. Strong encoding schema scores were obtained throughout the default mode network (*Figure 2B*; p<0.01 for all cortical ROIs). Additionally, there were strong schematic patterns in anterior but not posterior hippocampus (p<0.01 for whole hippocampus and anterior hippocampus; p=0.28 for posterior hippocampus).

### Reinstatement scores

To identify story-specific and schematic representations at recall, we measured the degree of neural reinstatement of each story during each recall period (*Figure 1D*). Here, we build on prior work on neural reinstatement (e.g., *Xue et al., 2010*; *Staresina et al., 2012*; *Ritchey et al., 2013*; *Wing et al., 2015*; *Tompary et al., 2016*; *Chen et al., 2017*) by using a hidden Markov model (HMM; *Baldassano et al., 2017*) to track reinstatement of sequences of patterns from the encoding phase.

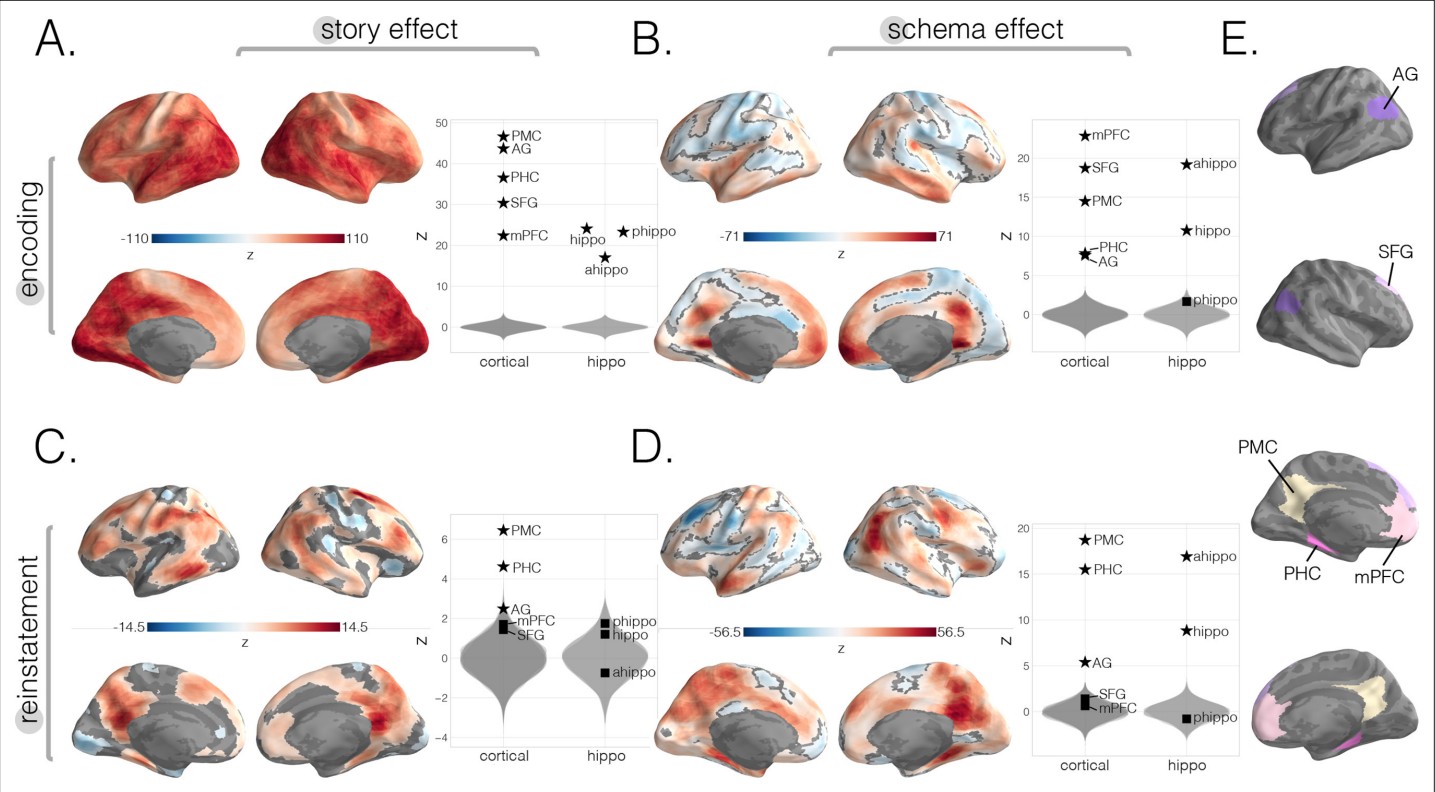

**Figure 2.** Neural story and schema strength during encoding and retrieval in whole-brain and specific cortical and hippocampal ROIs. (**A**) Encoding story scores. (**B**) Encoding schema scores. (**C**) Reinstatement story scores. (**D**) Reinstatement schema scores. All surface maps (**A–D**) were colorized with z-scores relative to the null distribution but thresholded via false discovery rate (FDR) correction for q < 0.05 after extracting p-values from a nonparametric permutation test. Plots depict effect sizes in ROIs, z-scored relative to the null distribution (gray). Starred points indicate significant differences after Bonferroni correction for multiple comparisons. (**E**) Locations of our cortical ROIs.

The online version of this article includes the following figure supplement(s) for figure 2:

**Figure supplement 1.** Neural story and schema strength during encoding and retrieval in whole-brain and specific cortical and hippocampal ROIs, computed within-subjects.

**Figure supplement 2.** Neural schema strength during encoding and retrieval in whole-brain and specific cortical and hippocampal ROIs, computed across-modality.

We first created 'encoding pattern templates' for each of the four events in each story by averaging the evoked response during encoding across all participants experiencing that event. We used these templates to construct 16 different HMMs (one for each story), where the states of each story-specific HMM corresponded to the sequence of four event patterns for that story during encoding. We then applied each of the 16 story-specific HMMs to each recall timeseries, to measure the degree to which each story's sequence of patterns was reinstated in that recall timeseries. Essentially, this HMM-fitting process involved – for a given story-specific HMM (from story A) and a given recall timeseries (from story B) – trying to model story B's recall timeseries under the assumption that it contained the same four 'template' event patterns (in the same order) as story A. The result of the HMM-fitting process was to subdivide the story B recall timeseries into four contiguous sections that best matched the four encoding event patterns from story A (see Materials and methods for more details). To measure neural reinstatement, we took the average neural patterns from each of these four sections of the story B recall timeseries and correlated these patterns with the actual encoding templates from story A (i.e., we correlated the part of the recall timeseries that the HMM matched to event 1 with the actual encoding pattern for event 1, likewise for events 2–4, and then we averaged these four correlations together). By the end of this process, each of the 16 story recalls for a given participant had been compared to each of the 16 story templates from the encoding period. Analogously to the encoding period, we computed – for each participant and each story – a *reinstatement story score* comparing

the reinstatement of the matching story to the reinstatement of other stories from the same schema, and a *reinstatement schema score* comparing the reinstatement of other stories from the same schema to the reinstatement of other stories from the other schema. These scores were computed in both searchlights and specific ROIs.

We found significant reinstatement story scores in regions overlapping with the default mode network (DMN), particularly lateral posterior SFG, central middle temporal gyrus, PHC, and AG with strongest effects in PMC (*Figure 2C*, q < 0.05). Our specific ROI analyses (*Figure 2C*) also showed strong reinstatement story scores in the same regions such as PMC (p<0.01), PHC (p<0.01), AG (p=0.04) but not did not show effects in mPFC (p=0.42), SFG (p>0.5), nor any of our hippocampal ROIs (full: p>0.5; anterior: p>0.5; posterior: p=0.29). For schema reinstatement (*Figure 2D*), the searchlight analysis revealed positive reinstatement schema scores in left anterior temporal pole (AT) as well as a negative effect in areas overlapping with left lateral SFG, indicating that stories from the same schema were more differentiated in this region (vs. stories from different schemas). Additionally, similar to our encoding results, our specific ROI analyses revealed strong schematic effects in anterior (p<0.01) but not posterior hippocampus (p>0.5). In contrast to our encoding results, we did not find schema reinstatement effects in mPFC (*Figure 2D*, p>0.5) nor SFG (p>0.5). However, we did find schema reinstatement effects in PMC (p<0.01), PHC (p<0.01), and AG (p<0.01).

## Predicting memory performance from story and schema encoding and reinstatement scores

To identify the degree to which story-specific or schematic neural representations predicted later memory for story details, we ran four separate leave-one-subject-out linear regressions using each of the four neural story and schema scores as single predictor variables (i.e., encoding story, encoding schema, reinstatement story, and reinstatement schema scores) and memory performance on individual stories (assessed as the number of story-specific details mentioned during free recall) as the outcome variable (*Figure 1E*). Note that the null distributions used to assess the statistical reliability of these regression results were constructed by scrambling the relationship between neural data and behavior within subjects (see Materials and methods for more details); as such, significant results indicate a reliable *within-subject predictive relationship* between neural measures associated with a story and behavioral recall performance for that story.

### Memory as a function of encoding story and schema scores

Encoding story scores predicted subsequent memory for story details in a very wide range of cortical regions (*Figure 3A*, q < 0.05). In agreement with the searchlight analysis, we also found significant positive effects in our cortical and hippocampal ROIs (*Figure 3A*, p<0.01 for all regions, except for p=0.03 for anterior hippocampus). The correlation between encoding story scores and subsequent memory was significantly more positive for posterior vs. anterior hippocampus (t(58) = –74.74, p<0.001). We found a sparser set of regions when using encoding schema scores to predict behavior. Based on our searchlight results, the strongest positive effects were found in regions overlapping with the left primary visual cortex, prostriata, anterior mPFC, entorhinal cortex, left posterior temporal sulcus, and left subcentral and postcentral gyrus (*Figure 3B*, q < 0.05). Interestingly, we also found reverse effects (with more schematic information at encoding predicting poorer story-specific memory performance) in multiple regions including bilateral SPL (*Figure 3B*, q < 0.05). When we looked for correlations between encoding schema scores and recall behavior in our cortical ROIs, we did not find any strong effects (*Figure 3B*), including our broad mPFC ROI, despite finding a correlation between encoding schema scores and recall behavior in its most anterior portion via the searchlight analysis. Lastly, when we analyzed subsections of the hippocampus, we found opposite correlations between encoding schema scores and subsequent memory, with significant negative effects in posterior hippocampus (p<0.01) and numerically positive but nonsignificant effects in anterior hippocampus (p=0.27). The effects in these two subregions were significantly different from each other when we compared their model coefficients (anterior – posterior) across participants (t(58) = 107, p<0.001).

### Memory as a function of reinstatement story and schema scores

Reinstatement story scores were related to recall of specific story details in many regions, with the strongest effects in areas overlapping with bilateral PMC, right mPFC, and right AT cortex (*Figure 3C*).

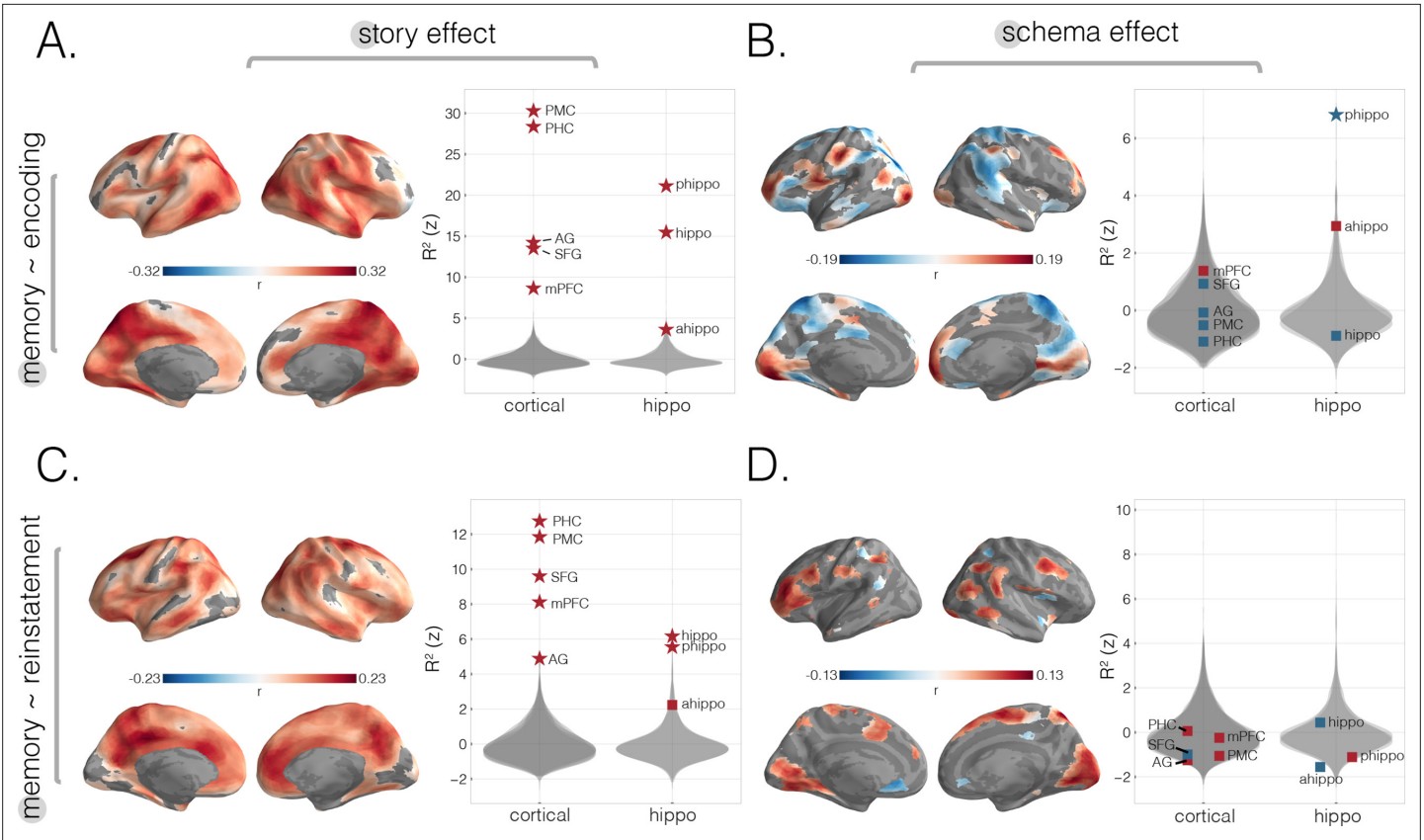

**Figure 3.** Predicting behavioral memory for story details with neural measures from encoding and recall. We predicted behavioral memory performance on held-out subjects based on each of our four neural scores (from **Figure 2**), across the neocortex and in specific ROIs. (**A**) Predicting memory scores using encoding story scores. (**B**) Predicting memory scores using encoding schema scores. (**C**) Predicting memory scores using reinstatement story scores. (**D**) Predicting memory scores using reinstatement schema scores. All surface maps were statistically thresholded by comparing model performance on held-out data to a null distribution and then FDR-correcting for q < 0.05. Surface maps are colored based on the correlation values between neural scores and behavioral memory performance. All violin plots show $R^2$ values describing model performance z-scored relative to the null distribution. Starred points indicate significant differences after Bonferroni correction for multiple comparisons. Point colors indicate directionality of prediction with red and blue for positive and negative associations, respectively.

The online version of this article includes the following figure supplement(s) for figure 3:

**Figure supplement 1.** Predicting posterior medial cortex (PMC) reinstatement story scores with encoding schema scores across cortex.

**Figure supplement 2.** Benefit of schema representation for detail memory.

**Figure supplement 3.** Predicting behavioral memory for story details with neural measures from encoding and recall within-subjects.

**Figure supplement 4.** Predicting behavioral memory for story details with across-modality neural schema measures from encoding and recall.

These effects were also confirmed in our larger cortical ROIs: There were significant effects in mPFC, SFG, and AG, and the strongest effects were in PMC and PHC (**Figure 3C**). In our hippocampus ROIs, we found that reinstatement story scores in posterior and not anterior hippocampus positively predicted subsequent memory (**Figure 3C**); the effect for posterior hippocampus was significantly larger than the effect for anterior hippocampus (t(58) = –27.45, p<0.001). In the searchlight analysis with reinstatement schema scores as a predictor variable, the strongest significant effects were in regions overlapping with bilateral primary visual cortices, bilateral posterior temporal sulcus, parietal regions, PHC, partial sections of medial SFG, right SPL, and lateral PFC (**Figure 3D**). There were no significant effects in the a priori cortical or hippocampal ROIs (**Figure 3D**). Additionally, because our PMC ROI was a strong predictor of story-specific behavioral memory and prior work implicates it in scene-specific representations (**Chen et al., 2017**), we wanted to determine how schematic representations across the brain at encoding relate to PMC's story-specific representations at recall (**Figure 3— figure supplement 1**). We found that, across the brain, schematic representations in bilateral visual cortex, AG, and fusiform cortex were the best predictors of PMC's reinstatement story effect.

## Intersection of significant schema effects and subsequent memory effects

To summarize the key regions in which schematic representations were robustly activated and supported memory performance, we intersected regions of the brain that showed significant schema scores and also showed a positive correlation with later memory for story details (*Figure 4A*). During encoding (*Figure 4A*), this conjunction analysis identified regions in visual cortex, left posterior temporal sulcus, prostriata, entorhinal cortex, left subcentral gyrus, postcentral sulcus, right lateral SFG, and anterior mPFC. For retrieval (*Figure 4A*), we found effects in visual cortex, posterior superior temporal sulcus, left fusiform gyrus, right SPL, right AG, PHC, medial SFG, and MFG.

To formally compare how schematic representations at encoding and retrieval predict subsequent memory differently, we ran a model coefficient comparison analysis in which we contrasted the encoding coefficients against the reinstatement coefficients (*Figure 4B*). This analysis corroborated our conjunction analysis (*Figure 4A*), revealing one set of regions that had significantly stronger relationships to subsequent memory at encoding than at recall (portions of left visual cortex, left posterior temporal sulcus, right prostriata, entorhinal cortex, left subcentral cortex, postcentral sulcus, frontal gyrus, and portions of anterior mPFC) and another set of regions that had significantly stronger relationships to subsequent memory at recall than encoding (visual cortex, SPL, clusters in posterior parietal regions, SFG, regions in frontal gyrus, and PHC).

## mPFC clustering and mediation analysis

### k-Means clustering

Do separate subregions within mPFC serve separate functions in memory? Given that our primary searchlight analysis revealed a relationship between encoding schema scores and subsequent memory in the most anterior part of mPFC, but this relationship was not significant in our whole-mPFC ROI (*Figure 3B*), we decided to perform a post-hoc clustering analysis to determine whether subregions of mPFC contributed to memory differently. Thus, in order to identify functional differences within mPFC, we ran a k-means clustering analysis. We first pooled the results of our eight whole-brain searchlight results together (i.e., *Figures 2 and 3*: story and schema encoding and reinstatement scores as well as their relationships to subsequent memory for story details) to obtain an eight-feature representation for each searchlight location (i.e., the eight features were the eight searchlight values for that location). We then ran a silhouette analysis on these eight-feature representations to determine the optimal number of clusters to use within a bilateral mPFC ROI mask (restricting the number of clusters k to be less than the number of input features). The analysis revealed that k = 2 yielded the highest average silhouette coefficient (s = 0.39). With this k = 2 solution, we found that the two clusters separated along the anterior-posterior axis in both hemispheres (*Figure 5A*). We then reran our previous analyses (e.g., encoding story score, encoding schema score, etc.) using these clusters as ROIs to identify (post-hoc) how the properties of these regions differed. We found that both clusters exhibited story and schema effects at encoding, but the contributions of these effects to subsequent memory differed across clusters: The encoding schema effect predicted subsequent memory in the anterior cluster but not the posterior cluster; by contrast, the relationship between the encoding story effect and subsequent memory was much larger in the posterior cluster than the anterior cluster. This flip in subsequent memory contributions between the anterior and posterior mPFC regions is consistent with a gradient of story representation to schema representation within mPFC, with schema representations in anterior (vs. posterior) mPFC being most critical for behavior.

### mPFC cluster mediation

Having shown that encoding schema scores in anterior mPFC predict subsequent behavioral recall performance, we sought to relate this effect to the neural reinstatement effects discussed earlier. One hypothesis is that schema information in anterior mPFC at encoding boosts behavioral recall by promoting the (subsequent) neural reinstatement of story-specific information in regions like PMC. To test this hypothesis, we looked at whether the relationship between encoding schema scores in the anterior mPFC cluster and behavioral recall was mediated by PMC story information at recall (*Figure 5B*). Indeed, we found that PMC story information acted as a partial mediator between mPFC_c0 schema information at encoding and later memory (indirect effect A * B = 0.029, 95% bias-corrected bootstrap CI [0.006, 0.056]).

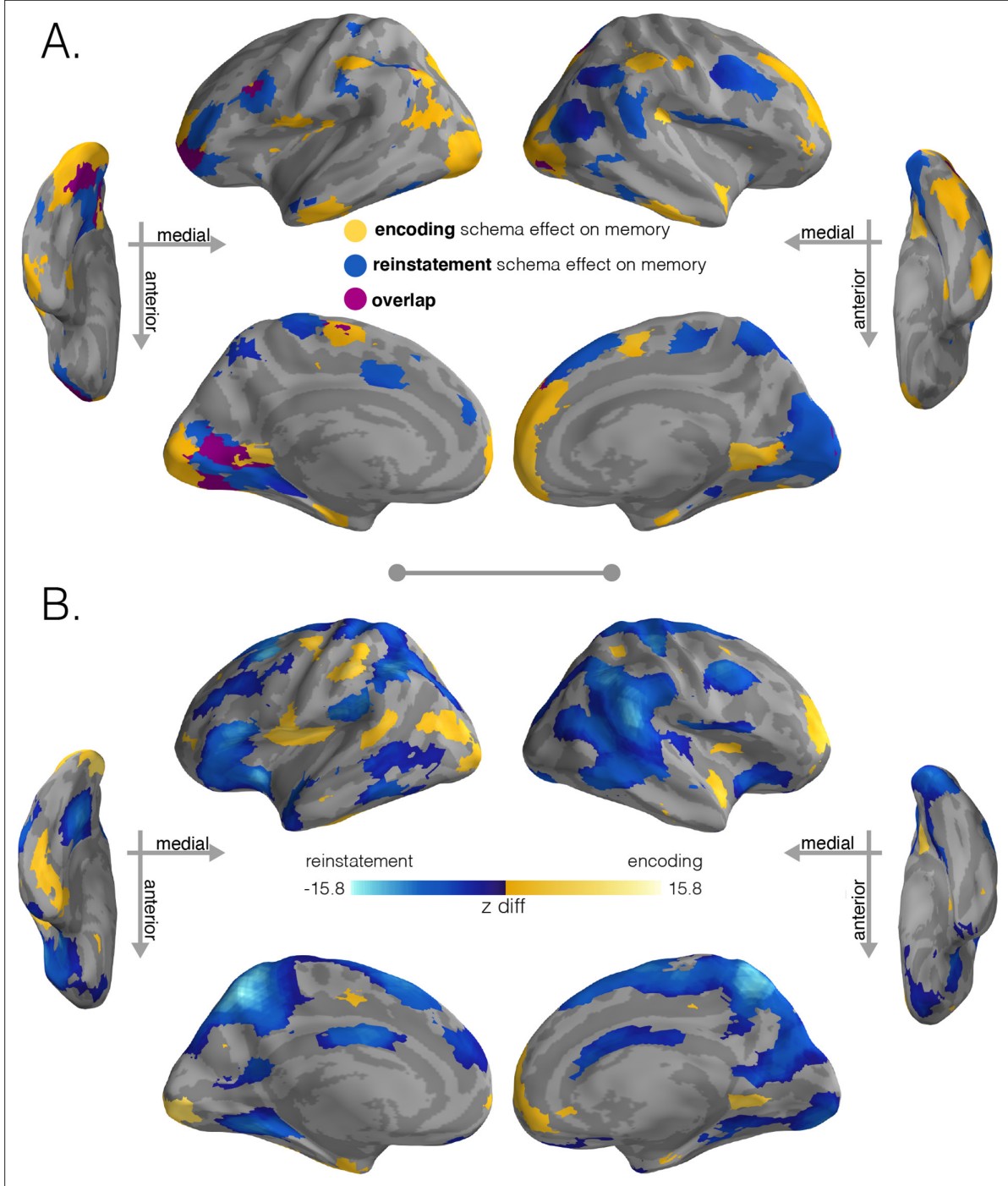

**Figure 4.** Regions with significant schema effects and positive associations with subsequent memory. (**A**) To combine the schema and regression effect at encoding, we intersected the regions showing a positive encoding schema effect (*Figure 2B*, q < 0.001) with the regions showing a positive relationship between the encoding schema effect and memory behavior (*Figure 3B*, q < 0.05); intersection shown in yellow. To combine the schema and regression effect at recall, we intersected the regions showing a positive reinstatement schema effect (*Figure 2D*, q < 0.001) with the regions showing a positive relationship between the reinstatement schema effect and memory behavior (*Figure 3D*, q < 0.05); intersection shown in light blue. Regions in purple indicate overlap between encoding and retrieval schema networks. (**B**) Model coefficient comparisons between schema effects on subsequent memory at encoding and recall. To further explore the regional specialization identified in (**A**), we ran a direct comparison between the model coefficients that predicted behavior (encoding schema coefficients - reinstatement schema coefficients). Stronger relationships between encoding schema scores and behavioral memory performance are shown in yellow, while stronger relationships between reinstatement schema scores and behavioral memory performance are shown in blue. Surface maps in (**B**) are colored based on the z-scored difference values (encoding – reinstatement) relative to a null distribution of differences. Significance was determined through nonparametric testing against a null distribution of differences after which p-values were converted to q-values with AFNI's FDR correction (q < 0.05).

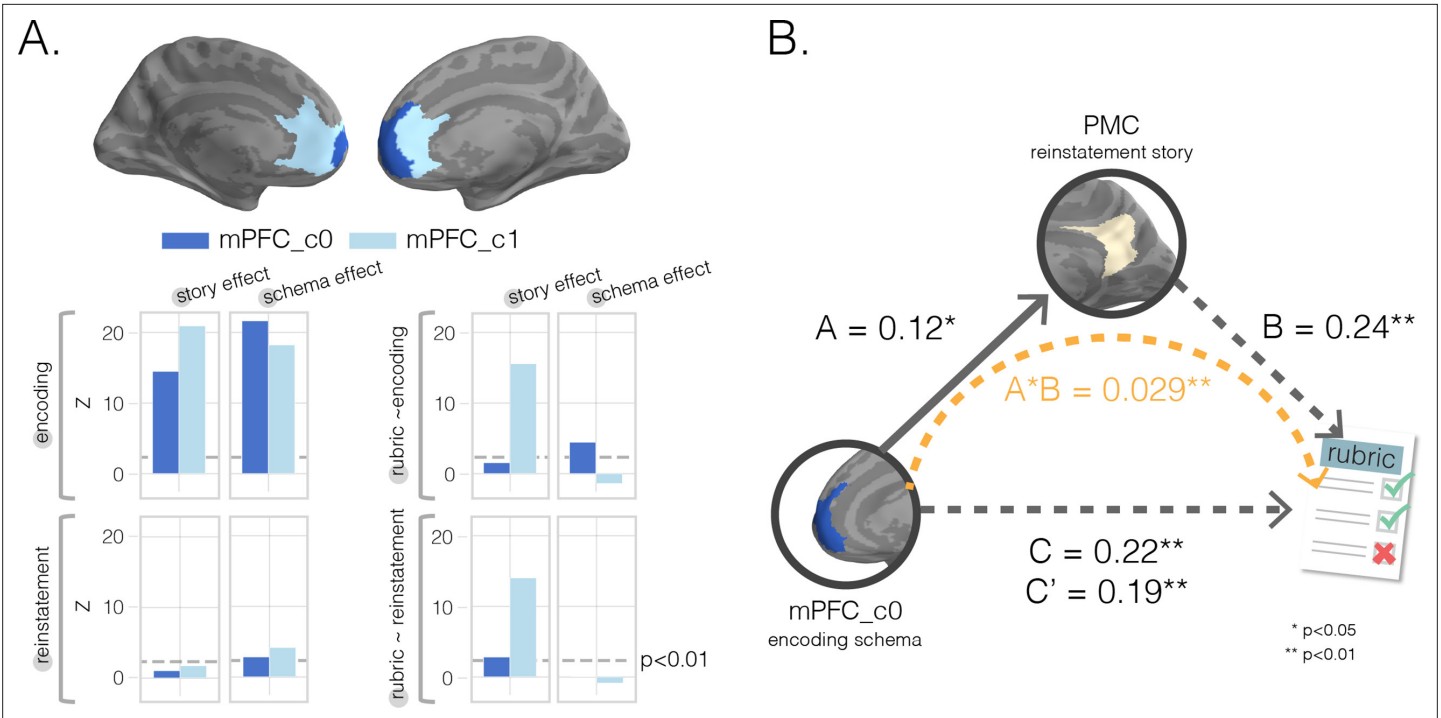

**Figure 5.** Mediation analysis with medial prefrontal cortex (mPFC) subclusters. (**A**) k-Means clustering results. We performed post-hoc k-means (k = 2) clustering within a bilateral mPFC ROI to identify subregions showing different profiles of results across our eight searchlight analyses; that is, each searchlight was assigned an eight-dimensional vector reflecting the scores assigned to that region in the eight searchlight analyses, and clustering was done on these eight-dimensional vectors (**Figures 2 and 3**) as features. Resulting (post-hoc) cluster effects are pictured in dark and light blue colors for anterior and posterior clusters, respectively. Consistent with our searchlight results, we find that encoding schema scores in voxels in the anterior cluster (c0) were associated with increased memory performance (top-right graph). (**B**) Mediation analysis. To determine whether posterior medial cortex (PMC) story information at recall mediated mPFC's impact on memory accuracy, we ran a single-mediator model with mPFC_c0's encoding schema score as the causal variable, PMC's reinstatement story score as the mediator, and rubric-derived memory scores as our outcome variable. We found that PMC reinstatement story scores were a significant partial mediator.

## Discussion

In this study, we investigated how our schematic knowledge about the sequential nature of common real-life experiences shapes memory for specific narratives during two distinct phases: when we initially encode a new experience, and when we search our memory to retrieve it. We examined regions in the brain that exhibited schematic patterns at encoding and retrieval and measured the degree to which schematic information in both of these stages predicted subsequent memory for story details. In our prior work using this airport/restaurant paradigm, we had identified a large region of mPFC that represents schema information at encoding. Here, we found that an (anterior) subset of this region had the property that the level of schema representation during encoding predicted subsequent memory for the story (measured using free recall). We also found that, while mPFC played an important role in schema representation during encoding of the stories presented here, it did not reliably represent schema information during recall of these stories, and the degree of schema representation in mPFC during recall did not reliably predict behavior. Consistent with ongoing research on functional differences along the long axis of the hippocampus (for a review, see **Poppenk et al., 2013**), we found a major difference in how schema representations in anterior and posterior hippocampus contributed to subsequent memory at encoding. Anterior hippocampus showed a high level of schema representation at encoding and a nonsignificant positive relationship between schema representation at encoding and subsequent memory; in contrast, the level of schema representation in posterior hippocampus at encoding was significantly *negatively* correlated with subsequent memory for the stories. Furthermore, neither hippocampal region showed significant relationships between schema representation and behavior at recall. More generally, the brain regions where schema representation during encoding predicted behavioral memory performance (visual cortex, left

posterior temporal sulcus, prostriata, entorhinal cortex, left subcentral gyrus, postcentral sulcus, right lateral SFG, and anterior mPFC) were surprisingly distinct from the brain regions where schema representation during recall predicted behavioral memory performance (bilateral visual regions that were generally more medial/anterior than the regions identified at encoding, posterior superior temporal sulcus, left fusiform gyrus, right SPL, right AG, PHC, medial SFG, and MFG). As a whole, these results provide evidence that event schemas support memory for the details of naturalistic narrative stimuli, and that the brain networks that provide this support are different when we are integrating situational information during perception and when we search for memories during retrieval.

Importantly, follow-up analyses revealed that, both at encoding and at recall, our ability to predict recall behavior based on neural schema scores and neural story scores (together) was better than prediction based on neural story scores alone (*Figure 3—figure supplement 2*). This implies that neural schema representations provide an extra benefit to recall, beyond what is obtained by merely representing story-specific details. The fact that different regions were involved at encoding and recall suggests that the mechanism by which schemas provide this extra benefit may be different at encoding and recall. As discussed in the Introduction, one way that activated event schemas can benefit encoding is by acting as a scaffold, providing a structured set of 'attachment points' for specific event details (e.g., *Bransford and Johnson, 1972*; *Alba and Hasher, 1983*; *Abbott et al., 1985*; *Tompary and Thompson-Schill, 2021*; *McClelland et al., 2020*); at retrieval, one way that schemas can benefit memory is by supporting a sequential memory cueing strategy, where participants step through the various stages of an event and ask themselves which specific details were associated with each stage (e.g., *Schank and Abelson, 1975*; *Anderson and Pichert, 1978*; *Bower et al., 1979*; *Alba and Hasher, 1983*; *Mandler, 2014*). The results of this study do not speak of the validity of either theory – more specialized designs will be needed to test these accounts.

## Stimuli and design

The schema literature in fMRI has been mostly split between studies that investigate the role of schemas at encoding and those that investigate their role in retrieval (but see *Bonasia et al., 2018*; *Sommer, 2017*; *van der Linden et al., 2017*; *Raykov et al., 2021*; *Reagh et al., 2021*). Of the studies focusing on the encoding phase, schemas have been operationalized by contrasting conditions in which participants have relevant prior knowledge vs. when they do not have this knowledge (*Maguire et al., 1999*; *van Kesteren et al., 2010a*; *van Kesteren et al., 2014*; *Raykov et al., 2018*; *Raykov et al., 2020*; *Keidel et al., 2018*; *Liu et al., 2017*; *Liu et al., 2018*; *Sommer, 2017*) or by using stimuli that are consistent vs. inconsistent with an activated schema (*van Kesteren et al., 2013*; *van Kesteren et al., 2020*; *Bonasia et al., 2018*; *van der Linden et al., 2017*). Of the studies focusing on the retrieval phase, schemas have been studied through spatial paired associate tasks (PAs) (*van Buuren et al., 2014*; *Sommer, 2017*; *Guo and Yang, 2020*; *Müller et al., 2020*), learned rules or hierarchies (*Wagner et al., 2015*; *Brod et al., 2015*), simple associations (*van Kesteren et al., 2010b*), static pictures (*Webb et al., 2016*; *Webb et al., 2019*; *van der Linden et al., 2017*), and short video clips (*Bonasia et al., 2018*; *Raykov et al., 2021*; *Reagh et al., 2021*). In contrast to this past work, our design employed naturalistic, temporally extended schema-consistent stimuli that were then paired with realistic unpaced verbal recall. Importantly, our design allowed us to neurally estimate the degree of story-specific and schematic representation for each individual story, at both encoding and retrieval, across cortex and also in hippocampus; we were able to leverage this to explore how all of these factors related to behavioral memory for story details in a story-by-story fashion.

## Relationship between schematic representations during encoding and subsequent memory

As was shown in a previous analysis of this dataset (*Baldassano et al., 2018*), schema representations were present at encoding in many regions previously identified in other studies of schemas, including mPFC (*van Kesteren et al., 2013*; *van Kesteren et al., 2020*; *van Kesteren et al., 2014*; *Raykov et al., 2020*; *Liu et al., 2017*; *Brod and Shing, 2018*; *Sommer, 2017*; *Bonasia et al., 2018*; *Reagh et al., 2021*), PMC (*Maguire et al., 1999*; *van Kesteren et al., 2013*; *Sommer, 2017*; *Bonasia et al., 2018*), SFG (*Bonasia et al., 2018*; *Brod and Shing, 2018*), PHC (*Keidel et al., 2018*; *van Kesteren et al., 2014*; *Liu et al., 2017*; *Bonasia et al., 2018*; *van der Linden et al., 2017*), AG (*Keidel et al., 2018*; *Bonasia et al., 2018*; *van der Linden et al., 2017*), and the hippocampus (*van Kesteren et al.,*

2013; *van Kesteren et al., 2014*; *Raykov et al., 2020*; *Liu et al., 2017*). We also identified strong schema representation in prostriata (*Mikellidou et al., 2017*), a region anterior to V1 and in between restrosplenial cortex (RSC) and PHC.

Since having similar patterns across multiple stories is partially in conflict with having highly distinct story-specific patterns, it is not immediately obvious that maintaining schematic patterns that are shared across stories should be helpful for remembering story-specific information (for evidence of a tradeoff between recall of item-specific vs. shared information, see *Tompary and Davachi, 2017*). However, we found multiple regions in which maintenance of this abstract schematic information was associated with improved memory for story details, including anterior mPFC, lateral frontal cortex, and portions of visual cortex (*Figure 4A*). Previous studies have shown that greater levels of mPFC activation at encoding are associated with better memory for schema-consistent stimuli (e.g., *Raykov et al., 2021*; *van Kesteren et al., 2013*; *van Kesteren et al., 2014*; *Brod and Shing, 2018*); our results extend the literature by revealing a within-subjects (across-story) relationship between the activation of anterior mPFC schema representations at encoding and memory for details of temporally extended naturalistic stimuli. These findings are compatible with the idea that mPFC might provide a schematic scaffold to which detailed representations are attached at encoding (*Gilboa and Marlatte, 2017*; *Schlichting and Preston, 2015*; *van Kesteren et al., 2012*). Surprisingly, we also found that schema representations in visual cortex contribute to memory. It is possible that certain visual features (e.g., visual features of security lines for airports, or tables for restaurants) are central to the mental representations of these airport and restaurant schemas; as such, increased attention to these visual features (for movies) and/or visualization of these features (for audio narratives) may reflect stronger schema representation, leading to improved memory encoding.

In the hippocampus, we found diverging effects in posterior and anterior subregions, with schematic patterns at encoding being nonsignificantly helpful for memory in anterior hippocampus but significantly harmful to memory in posterior hippocampus (the difference between these effects was also significant). Furthermore, while both posterior and anterior hippocampus exhibited significant story-specific representation at encoding, the correlation between encoding story scores and subsequent memory was significantly larger for posterior hippocampus. Taken together, these results suggest that posterior hippocampus plays an especially important role in representing story-specific details, consistent with theories of gist vs. detail representations in the hippocampus (*Guo and Yang, 2020*; *Audrain and McAndrews, 2020*; *Poppenk et al., 2013*; *Brunec et al., 2018*; *Schlichting et al., 2015*; *Collin et al., 2015*; *Sekeres et al., 2018*; for data suggesting a reversed gradient, see *Tompary and Davachi, 2017*; *Dandolo and Schwabe, 2018*).

## Relationship between schematic representations during retrieval and memory performance

We identified a set of regions in which schematic codes were reactivated during retrieval, and the degree of reactivation was related to behavioral recall performance. These regions were largely nonoverlapping with those from the encoding-phase analysis and included fusiform gyrus, MFG, and posterior parietal regions including right AG and SPL. Left fusiform gyrus and AG have been associated with visual imagery (*Spagna et al., 2021*; *Ragni et al., 2020*; *Kuhl and Chun, 2014*), and posterior parietal regions such as SPL have been implicated in top-down attention during episodic memory retrieval (*Hutchinson et al., 2014*; *Wagner et al., 2005*; *Cabeza et al., 2008*) and general memory success (*Brod et al., 2015*; *Webb et al., 2016*). Because no visual cue (apart from the title of a story) was provided during recall, participants may need to rely on top-down generation of visual cues to orient to particular schema stages (e.g., generating a mental image of what airport security usually looks like, to cue memory for the airport security part of an airport narrative).

Many of the regions listed above (posterior parietal regions as well as lateral temporal cortex, SFG, MFG, and visual regions) have previously been implicated in schematic memory (*Guo and Yang, 2020*; *Webb et al., 2019*; *Brod et al., 2015*; *van der Linden et al., 2017*), but they have also been reported to be involved in memory even when there is no schema manipulation (*van Buuren et al., 2014*; *van Kesteren et al., 2020*; *van Kesteren et al., 2010b*; *Webb et al., 2016*; *Webb et al., 2019*; *Brod et al., 2015*). Since our study can separately measure both story-specific and schematic reactivation patterns during naturalistic recall, we were able to show that there was a memory boost from schema-related reactivation in these regions in addition to more general story reactivation effects.

Given the strong involvement of mPFC during schematic encoding, it has been hypothesized that mPFC may play a role at retrieval by providing schematic cues for memory search (*van Kesteren et al., 2012*). While some studies have found that schema-related activity in mPFC during retrieval benefits memory (*Brod et al., 2015*; *van Kesteren et al., 2010b*; *Müller et al., 2020*; *Webb et al., 2019*; *Raykov et al., 2021*), others have not (*van Buuren et al., 2014*; *Webb et al., 2016*; *Guo and Yang, 2020*; *van der Linden et al., 2017*; *Reagh et al., 2021*). In our study, we did not observe strong mPFC schema reinstatement, nor were we able to relate it to a behavioral memory benefit. It is possible that schema representations in mPFC contribute to retrieval, but we failed to detect this contribution, for example, because they only emerge at specific time points during recall, or they only arise after sleep consolidation (*van der Linden et al., 2017*; *Brod et al., 2015*; *van Kesteren et al., 2010b*), or they contribute through interactions with other brain regions (*Guo and Yang, 2020*; *van Kesteren et al., 2010b*). Alternatively, schematic representations in mPFC during recall may be associated less with accurate recall of specific story details and more with verbal descriptions of schematic elements of the narrative. Because the rubric we made for scoring memory performance tracks recall of story-specific details, it is not ideal for measuring the extent to which a recall conforms to the general (restaurant or airport) schema. Future work exploring the relationship between neural measures and verbal recall of schematic features could further deepen our understanding of the correspondence between the brain and behavior.

## Within-subject and across-modality supplementary results

*Baldassano et al., 2018* found that schema representation at encoding across many of the same regions reported here (*Figure 2*) was modality-independent. We therefore introduced a variant of our main analysis in which we contrasted schematic representations across-modality (i.e., video story vs. audio stories; *Figure 2—figure supplement 2*) and found that this contrast reproduced results previously reported (*Baldassano et al., 2018*) and corroborated the results reported in the main text. When we related these across-modality neural schema scores to behavior (*Figure 3—figure supplement 4*), we again found positive relationships to memory performance in mPFC at encoding; among a sparser set of regions at recall, we also found a positive relationship to memory in anterior hippocampus, suggesting that hippocampus can generalize across both content (i.e., shared structure across stories) and the surface form of content (i.e., modality).

Prior work has also shown that between-subject analyses can improve signal-to-noise ratios while revealing shared representations across people (*Baldassano et al., 2018*; *Baldassano et al., 2017*; *Chen et al., 2017*; *Zadbood et al., 2017*; *Simony et al., 2016*). We therefore ran another analysis variant in which we looked for within-subject effects (*Figure 2—figure supplement 1*, *Figure 3—figure supplement 3*), the results of these within-subject analyses were generally consistent with the between-subject results reported in the main text, with a few exceptions. First, in contrast to the between-subject results, there was no significant relationship between mPFC encoding schema scores and behavior. Also, the negative relationship between encoding schema scores in posterior hippocampus and subsequent memory was no longer significant (but the positive relationship between reinstatement schema scores in posterior hippocampus and subsequent memory was still significant).

## Conclusion

In our study, we derived neural measures of story-specific and schematic representations in the brain during the perception and recall of narratives conforming to naturalistic event schemas. Our results extend the literature on the benefits of schemas for memory performance, relating the maintenance of schematic representations to a continuous behavioral measure of detailed memory for realistic narrative stimuli. We found converging support for the idea that schema representations in mPFC play an important role in memory encoding, but also striking differences between regions where schema representation at encoding was useful for memory, and regions where schema representation at retrieval was useful for memory. These findings can serve as a foundation for future work that seeks to further delineate the contributions of these encoding-specific and retrieval-specific schema networks.

# Materials and methods

## Participants

Data were collected from a total of 31 participants between the ages of 18–34 (15 females, 16 males). The perception (movie-watching and story-listening) data from these participants have been previously reported (*Baldassano et al., 2018*). At the end of the study, participants were paid and debriefed about the purpose of the study. Every effort was made to recruit an equal number of female and male participants and to ensure that minorities were represented in proportion to the composition of the local community. The experimental protocol was approved by the Institutional Review Board (IRB) of Princeton University, and all participants provided their written informed consent (IRB #7225). Due to data loss during the recall phase, one participant (female) was excluded from the recall analyses.

## Stimuli

The stimuli were designed to conform to two naturalistic schematic scripts that participants had encountered throughout their lifetimes. Each of the 16 stories described the schematic script of either eating at a restaurant or catching a flight at an airport (*Bower et al., 1979*). Each narrative was written or edited to follow a specific four-stage event structure. For restaurant stories, the event structure consisted of (1) entering and being taken to a table, (2) sitting with menus, (3) ordering food and waiting for its arrival, and (4) food arriving and being eaten; while airport narratives consisted of (1) entering the airport, (2) going through the security, (3) walking to and waiting at the gate, and (4) getting on board the plane and sitting in a seat.

The videos were movie clips sampled from films (restaurant: *Brazil, Derek, Mr. Bean, Pulp Fiction*; airport: *Due Date, Good Luck Chuck, Knight and Day, Non-stop*) that were edited for length and to conform to the four-stage script. The audio stimuli were adapted from film scripts (restaurant: *The Big Bang Theory, The Santa Clause, Shame, My Cousin Vinny*; airport: *Friends, How I Met Your Mother, Seinfeld, Up in the Air*) that were also edited for length and to match the schematic script. All audio narratives were read by the same professional actor. Each story, whether video or audio, was approximately 3 min long.

## Data acquisition and preprocessing

Data were acquired with a voxel size of 2.0 mm isotropic and a TR of 1.5 s (see *Baldassano et al., 2018*, for a full description of the sequence parameters). After fMRI data were aligned and preprocessed to correct for B0 distortion and fsaverage6 resampling, the resampled data were further preprocessed with a custom Python script that first removed nuisance regressors (the 6 degrees of freedom motion correction estimates, and low-order Legendre drift polynomials up to order [1 + duration/150] as in Analysis of Functional NeuroImages [AFNI]) (*Cox, 1996*), then z-scored each run, and then divided the runs into the portions corresponding to each stimulus (see *Baldassano et al., 2018*, for a more detailed description of our preprocessing pipeline).

## Experimental paradigm

After listening to a short unrelated audio clip to verify that the volume level was set correctly, participants were presented with four encoding runs, using PsychoPy (RRID:SCR_006571; *Peirce, 2007*). Each run consisted of interleaved video and audio stories, with one story from each modality and schema in each run, and a randomized run order across subjects. Every story was preceded by a 5 s black screen followed by a 5 s countdown video. The title of each story was displayed at the top of the screen throughout the story (the screen was otherwise black for the audio narratives). Participants were informed that they would be asked to freely recall the stories after all 16 had been presented.

During the recall phase, participants were asked to freely verbally recall (at their own pace) the details of each story when cued by the title of the story-to-remember. When participants finished recalling a particular story, they said "Done" to signal the experimenter for the next title. There were four recall runs in total. During each recall run, participants were cued to recall four stories, with a 1 min rest between each story recall. After recalling all 16 stories, while still being scanned, participants were asked to provide verbal descriptions of the typical experience of eating at a restaurant and the typical experience of going through an airport.

## Searchlights and ROIs

### Searchlights

Our searchlights were generated by randomly sampling a center vertex of the fsaverage6 surface mesh and identifying all vertices within 11 steps from it. Because the vertex spacing within the fsaverage6 mesh is 1.4 mm, the resulting radius is 15 mm. Searchlights were repeatedly sampled (discarding searchlights containing fewer than 100 vertices with valid timeseries) until every center vertex was included in at least 10 searchlights. This process yielded 1483 searchlights per hemisphere.

### A priori ROIs

Following recent work on the encoding of narrative event schemas using the same encoding dataset (*Baldassano et al., 2018*), as well as prior research on the representation of high-level situation models (*Zadbood et al., 2017*; *Chen et al., 2017*; *Baldassano et al., 2017*; *Kurby and Zacks, 2008*; *Radvansky and Zacks, 2017*; *Nguyen et al., 2019*; *Clewett et al., 2019*), we focused our main ROI analyses on mPFC, PMC, SFG, AG, and PHC because of their consistent presence in naturalistic paradigms and their role in maintaining schema representations during encoding. The regions were extracted from an established 17-network atlas on the fsaverage6 surface (*Thomas Yeo et al., 2011*) that formed part of the larger default mode network. Our full hippocampus ROI was extracted from a freesurfer subcortical parcellation, which was then further split between an anterior at y > –20 and posterior portion at y <= –20 in MNI space (*Guo and Yang, 2020*; *Poppenk et al., 2013*).

## Measuring story and schema strength in verbal and neural data

### Encoding similarity matrix

For each story, we created four regressors to model the neural response to each of the four schematic events (i.e., the four stages of the script), with an additional nuisance regressor to model the initial countdown. The four regressors (and nuisance regressor) in our design matrix were placed temporally by using hand-labeled timestamps that marked event transitions in the narratives. These were convolved with a hemodynamic response function (HRF) from AFNI (*Cox, 1996*) and then z-scored. We extracted the characteristic spatial pattern across vertices for each schematic event within a story by fitting a general linear model (GLM; within each participant) to the timeseries of each vertex using these regressors. Next, to quantify the degree to which stories evoked similar neural patterns, we used intersubject spatial pattern similarity (e.g., *Raykov et al., 2020*; *Baldassano et al., 2018*; *Chen et al., 2017*) – specifically, the event-specific patterns for a given story/participant were always compared to patterns that were derived from the N-1 other participants (by averaging the timecourses for the N-1 other participants for a given story, and then fitting a GLM to that averaged timecourse to identify the four event-specific patterns for that story). To compute the similarity for a given pair of stories (call them A and B), the pattern vectors for each of story A's four events were correlated with the pattern vectors for each of story B's four events (i.e., the event 1 pattern for story A was correlated with the event 1 pattern for story B from the N-1 other participants; the event 2 pattern for story A was correlated with the event 2 pattern for story B from the N-1 other participants, etc.). These four correlation values for a given pair of stories (event 1 to event 1, event 2 to event 2, etc.) were averaged into a single value. For each participant, this sequence of steps was used to compare that participant's representation of each story to the N-1 other participants' representation of each story. The net result of this process was a 16 × 16 correlation (similarity) matrix for every participant, containing the (intersubject) neural similarity of each story to every other story (see *Figure 1C*).

Because prior work has shown that between-participant analyses can uncover shared neural event representations across participants (*Baldassano et al., 2018*; *Baldassano et al., 2017*; *Chen et al., 2017*; *Zadbood et al., 2017*) and boost signal-to-noise ratio (*Simony et al., 2016*), we used an across-subject encoding similarity matrix as our default analysis. However, we also included a within-subject encoding similarity matrix for comparison purposes. To create this within-subject similarity matrix, we followed the same steps described in the previous paragraph, but – rather than comparing a participant's event-specific patterns of a particular story against the average patterns derived from the other N-1 participants – we compared that participant's event patterns for a particular story against their own event-specific patterns for every other story. This results in a 16 × 16 symmetric correlation matrix with valid off-diagonal entries, corresponding to the event-specific similarity of one story to another,

and ones in the principal diagonal, corresponding to the perfect similarity of a story's patterns with itself.

## Reinstatement similarity matrix

First, we created a template for each of the four events in each story by using a GLM to extract the multivoxel BOLD pattern for that event within each participant, and then averaging across participants to get a single spatial pattern for that event (*Figure 1D*). We then sought to measure the extent to which these story-specific patterns were reinstated during the free recall period with the HMM approach used in *Baldassano et al., 2017* and *Baldassano et al., 2018*. The model makes the assumption that, when recalling a story, the event pattern templates from encoding are replayed in the same ordered sequence. The variance parameter for the model was calculated per participant by measuring the variance of that story's mean event patterns at perception. Given a template pattern for a story (i.e., its four-stage encoding pattern), and the timeseries for the recall of that story, the model computes a probability that each time point of the recall belongs to each of the four template events. We then computed a weighted average spatial pattern for each event during the recall using the probability matrix as the weights. To determine the strength of reinstatement between the template story and the recalled story, each of the four encoding event template patterns was correlated with all of the four recall event patterns, the strength of reinstatement was measured as the difference between the correlations for corresponding (e.g., encoding template event 1 and recall event 1) and noncorresponding events (e.g., encoding template event 1 and recall event 2). This difference measure per event was averaged and was repeated for all combinations of template story and recall story, yielding a 16 × 16 encoding-recall similarity matrix per participant (see *Figure 1D*). Importantly, while our HMM method is biased to recover patterns that match the encoding templates, this bias applies equally regardless of which stories are being compared; our reinstatement story and schema measures control for this bias by looking at the *relative* degree of reinstatement across different comparisons (e.g., comparing reinstatement of stories from the same schema vs. stories from the other schema).

Just as we created a within-subjects version of our encoding similarity matrix, we also created a within-subjects version of our reinstatement similarity matrix. To do this, we followed the same steps described in the previous paragraph with the exception that we skipped the averaging-across-participants step. Specifically, an encoding template for each of the four events in each story was made for each participant by using a GLM to extract the multivoxel BOLD pattern for each event within each participant. Rather than averaging these event patterns across participants to get a single four-event template for each story, each participant had their own four-event story template, resulting in 16 unique story templates per participant. Then, just as before, to determine the degree to which story-specific patterns (i.e., encoding templates) were reinstated at recall, but this time in a within-subjects approach, we used an HMM, as previously described, to calculate the strength of reinstatement for the encoding patterns of a particular story within their corresponding free recall of that story and every other story. This yielded a 16 × 16 (within-subject) encoding-recall similarity matrix per participant.

## Story and schema scores

Because we generated both an encoding and reinstatement similarity matrix for every participant, we could then perform contrasts for each stimulus for each participant (during encoding or recall) to measure the extent to which neural representations contain story-specific or schematic information (*Figure 2*).

*Story score* (*Figure 1C and D*): To compute the *story score* for a particular story, we contrasted that story's similarity to itself (a square on the diagonal of the similarity matrix) with the average of that story's similarity to other stories from the same schema and modality (the modality restriction was done to avoid effects driven by overall modality differences unrelated to this particular story). We determined statistical significance for the difference in similarity using a nonparametric permutation test in which we randomly permuted the stories within a schema (and within modality) 1000 times to generate a null distribution of differences. A p-value was computed as the proportion of times a difference in the null distribution was greater than or equal to the difference of the correctly labeled data.

*Schema score* (**Figure 1C and D**): To compute the *schema score* for a particular story, we contrasted the average of that story's similarity to other stories from the same schema with the average of that story's similarity to other stories from the other schema (using only stories from the same modality). Statistical significance was determined in a nonparametric permutation test in which schema labels of stories (within the same modality) were randomly permuted 1000 times.

For completeness, we also computed an across-modality schema score for each story by contrasting the average of that story's similarity to other stories from the same schema and different modality with the average of that story's similarity to other stories from the other schema and different modality. This means that for a particular *video* story we contrasted the average similarity of that story to *audio* stories from the same schema against the similarity of that story to *audio* stories from the other schema, and vice versa. Statistical significance was determined as described in the previous paragraph.

Note that, for the within-subjects *encoding* similarity matrix described earlier, a story score cannot be computed because the similarity of a participant's encoding template to itself is always going to be 1 (**Figure 2—figure supplement 1**). Therefore, for the within-subjects encoding similarity matrix, only a schema score was calculated, but for the within-subjects reinstatement similarity matrix, both story and schema scores were computed.

To generate brainmaps of these scores, story and schema information was extracted from the encoding and reinstatement similarity matrices computed at each searchlight (**Figure 1**). To convert searchlights back to the cortical surface, the score for each vertex was computed as the average scores of all searchlights that included that vertex. Similarly, we averaged the null distributions for all searchlights that included a vertex to get a single null distribution per vertex. p-Values were obtained per vertex through a two-sided nonparametric permutation test that looked for the proportion of times an absolute value in the null distribution (created by shuffling story labels separately for each participant) was greater than the absolute value of the original averaged story or schema score. We then converted these p-values to q-values using the false discovery rate correction from AFNI (**Cox, 1996**).

## Behavior

*Verbal recall analysis* (*rubrics*; **Figure 1E**): Hand-scored rubrics were used to provide a quantitative behavioral measure of memory recall performance for details within a story (available here). Rubrics for videos included points for recalling unchanging ('static') details (e.g., character appearance, set design) and 'dynamic' details (e.g., combined dialogue and visual descriptions). For audio stories, only 'dynamic' details were tracked (given the lack of visual information). Transcripts for audio stories were split into sentences and points were awarded if a detail from at least a fraction of a sentence was recalled. Participant audio was recorded during free recalls and was manually timestamped, transcribed, and scored for memory performance using the rubrics. Memory performance was measured by the number of details remembered (sum of points) and normalized by total possible details for a given story (as measured by max possible rubric score for a story). Two independent coders scored every participant's memory performance (intercoder reliability, Pearson $r = 0.95$), and final scores per story were averaged across both independent coders.

## Predicting behavioral performance from neural scores

We next wanted to identify whether the story and schema scores at encoding or recall predicted behavioral memory performance (**Figure 3**). In other words, how does the neural representation of story and schema information at either encoding or recall predict later memory?

To answer this question, we conducted four separate leave-one-participant-out linear regression analyses for each ROI or searchlight. Each of the four regression analyses used a particular neural score (either encoding story, encoding schema, reinstatement story, or reinstatement schema) to predict behavioral recall performance on a story-by-story basis. The regression models were trained on neural scores and behavioral scores from all but one participant; we then used the trained model to predict the left-out participant's 16 behavioral recall scores (one per story; **Figure 1F**) based on that participant's neural scores. Each of the four regressions was run with each of the 30 subjects as a test subject, providing a 30 × 16 matrix of behavioral predictions on held-out subjects. With these predictions, model performance was measured by variance explained ($R^2$) compared to a baseline model of simply predicting the average rubric score of the N-1 group. Statistical significance was

determined through nonparametric permutation testing, in which a null distribution of 1000 values was made by shuffling the story scores within each subject (thereby keeping the subjects intact) before running the leave-one-out regression. Given this type of permutation, any significant results observed are due to within-subject (but across-stories) relationships between neural measures and behavior. Using a mixed-effects model could potentially increase the sensitivity of our regression analysis by removing across-subject variance when measuring the relationship between neural and behavioral measures. However, since our analysis requires fitting this model for each searchlight, permutation, and left-out subject, it was computationally infeasible to fit this kind of model. Note that our choice to use a simpler model should lead only to false negatives (reducing sensitivity to subtle effects) and not to false positives in our results. To visualize searchlight results on the cortical surface, we averaged $R^2$ scores across searchlights in the same way that was described above for the story and schema scores (i.e., each vertex was assigned the average $R^2$ across all of the searchlights that included that vertex).

With simple linear regression, predictions of rubric scores below zero were possible, despite zero being the lowest possible rubric score. To enforce realistic predictions of rubric scores of greater or equal to zero, we also ran the same regression procedure with a logistic output layer; the results of this analysis were highly similar to the results that we obtained when we used linear regression. Consequently, for the sake of simplicity, we only describe the results for simple linear regression here.

These same analyses were rerun using story and schema scores derived using a within-subjects approach and also using the across-modality schema scores described previously.

## ROI to ROI correlations

Because story information was most strongly reinstated in PMC (*Figure 2*), and this reinstatement was highly predictive of behavioral rubric scores (*Figure 3*), we examined whether there were neural signals during encoding that were predictive of later PMC story reinstatement. To do this, we ran a linear regression with PMC's reinstatement story score as the dependent variable and each searchlight's encoding schema score as the independent measure. To test for significance, we generated a null distribution in which story labels for the dependent variable were shuffled within subjects. In this nonparametric permutation two-sided test, p-values were computed by calculating the proportion of absolute values above the test value. Searchlights were converted back to vertex space before converting the p-values to q-values with AFNI's FDR correction. To visualize the results on a brainmap, vertices were thresholded at q < 0.05.

## Schema representation and subsequent behavior

To identify regions where schema information was represented and the degree of that schema representation influenced memory, we intersected positive schema effects (thresholded by FDR < 0.001; *Figure 2B* or *Figure 2D*) with regions that were positively correlated with later memory (thresholded by FDR < 0.05; *Figure 3B* or *Figure 3D*). We did this for encoding and recall separately (*Figure 4A*).

To quantify how schema representations influenced memory at both encoding and recall (*Figure 4B*), we compared their model coefficients (encoding – reinstatement). Statistical significance was computed via two-sided nonparametric testing of a null distribution of differences between model coefficients at encoding vs. reinstatement. p-Values were then converted to q-values with AFNI's FDR correction.

## Benefit of schema representation for detail memory

In order to determine the degree to which neural schema scores at encoding or retrieval improved prediction of behavioral memory for story-specific information (beyond what could be predicted from neural story scores alone), we compared a single-regressor baseline model with story scores as the sole predictor against a nested model with schema scores as an additional predictor (*Figure 3— figure supplement 2*). We computed an F-statistic to measure the degree to which the nested model, with the additional model term, better explains the data. To test for statistical significance, we ran a nonparametric permutation test, in which a null distribution of F-statistics was generated for 1000 model comparisons after shuffling labels. Extracted p-values were converted to q-values using AFNI's FDR correction.

## k-Means clustering and mediation analysis

Although the full mPFC ROI showed strong schema representation during perception (*Figure 2*), the behavioral prediction searchlights revealed that schema information only predicted behavior in the most anterior portion of mPFC (*Figure 3B*). To explore the differential functional roles of mPFC subregions, we ran a post-hoc k-means clustering analysis to segment our mPFC ROI into two clusters with distinct functional profiles (*Figure 5*). Using the results of eight different searchlights (*Figures 2 and 3*) as features, we generated clusters across multiple k's bilaterally on searchlight vertices using our a priori mPFC ROI as a mask. We first ran a silhouette analysis to determine the optimal k (restricting k to be less than the number of input features); then, for each resulting cluster, we calculated new similarity matrices, extracted story and schema scores, and ran our behavioral prediction analysis (*Figure 5A*).

We also ran an additional mediation analysis to identify the extent to which PMC and mPFC subregions interacted to support recall (*Figure 5B*). Our goal was to determine whether the behavioral impact of schematic representations in mPFC at encoding was mediated through the reinstatement of story information in PMC. To do this, we ran a traditional single-mediator model in which the causal, mediator, and outcome variables were mPFC subregion schema information at encoding, PMC story information at recall, and rubric scores, respectively (*Baron and Kenny, 1986*). The total effect of the causal variable mPFC schema at encoding on the outcome rubric scores (path c) was calculated by running a linear regression with each regressor standardized. The significance of the effect was computed by generating a null distribution from shuffling the labels of the outcome variable, generating a corresponding z-value for the original effect, and converting to a p-value from the survival function of the normal distribution. This same procedure was used to test for the significance of each individual component in the indirect effect (paths a and b) as well as the direct effect (path c'). To test for statistical significance of the indirect effect (i.e., mediated effect), we performed a bias-corrected bootstrap test (*Efron and Tibshirani, 1994*). To determine the specificity of this effect, we also ran a variant of this analysis where we swapped the roles of the two ROIs using the PMC encoding schema score as the causal variable and mPFC subregion reinstatement story score as the mediator. We found no significant effects in this analysis.

## Data and code

All data is openly available for download at OpenNeuro.

Scripts used for analysis also available at GitHub, (copy archived at swh:1:rev:af3cf26ee9a82f-fa898708ecc3c2cfdfd9bd6633; *Masís-Obando, 2021*).

## Acknowledgements

We would thank Uri Hasson for help in the design and analysis of this study; Sophia Africk for help in grading the rubrics; Samantha Cohen, Alex Barnett, and Silvy Collin for comments and suggestions on the manuscript; Norbert J Cruz-Lebrón, Jamal Williams, Sam Nastase, and Meir Meshulam for insightful conversations; and the Norman, Hasson, and Baldassano labs for their general comments and support. This work was supported by NIH R01MH112357 to KAN and the NINDS D-SPAN award F99 NS120644-01 to RMO.

## Additional information

### Funding

| Funder | Grant reference number | Author |
| --- | --- | --- |
| National Institutes of Health | R01MH112357 | Kenneth A Norman |
| National Institute of Neurological Disorders and Stroke | F99 NS120644-01 | Rolando Masís-Obando |

The funders had no role in study design, data collection and interpretation, or the decision to submit the work for publication.

## Author contributions
Rolando Masís-Obando, Conceptualization, Data curation, Formal analysis, Funding acquisition, Methodology, Visualization, Writing - original draft, Writing - review and editing; Kenneth A Norman, Conceptualization, Funding acquisition, Investigation, Methodology, Project administration, Supervision, Writing - review and editing; Christopher Baldassano, Conceptualization, Investigation, Methodology, Project administration, Software, Supervision, Writing - review and editing

## Author ORCIDs
Rolando Masís-Obando (ID) http://orcid.org/0000-0001-7382-3778
Kenneth A Norman (ID) http://orcid.org/0000-0002-5887-9682
Christopher Baldassano (ID) http://orcid.org/0000-0003-3540-5019

## Ethics
The experimental protocol was approved by the Institutional Review Board (IRB) of Princeton University and all participants provided their written informed consent (IRB Protocol number 7225).

## Decision letter and Author response
Decision letter https://doi.org/10.7554/eLife.70445.sa1
Author response https://doi.org/10.7554/eLife.70445.sa2

## Additional files

### Supplementary files
• Transparent reporting form

### Data availability
fMRI data for all portions this study are publicly available in OpenNeuro under accession code ds001510. All code used for analysis available on GitHub (copy archived at swh:1:rev:af3cf26ee9a82ffa898708ecc3c2cfdfd9bd6633).

The following dataset was generated:

| Author(s) | Year | Dataset title | Dataset URL | Database and Identifier |
|---|---|---|---|---|
| Baldassano C, Masís-Obando R, Hasson U, Norman KA | 2021 | Schematic narrative perception and recall (intact) | https://doi.org/10.18112/openneuro.ds001510.v2.0.2 | OpenNeuro, 10.18112/openneuro.ds001510.v2.0.2 |

The following previously published dataset was used:

| Author(s) | Year | Dataset title | Dataset URL | Database and Identifier |
|---|---|---|---|---|
| Baldassano C, Masís-Obando R, Hasson U, Norman KA | 2020 | Schematic narrative perception and recall (intact) | https://doi.org/10.18112/openneuro.ds001510.v1.0.4 | Openneuro, 10.18112/openneuro.ds001510.v1.0.4 |

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
