## [Editor Report]

This paper reports a methodologically rigorous investigation into the neural mechanisms supporting encoding and retrieval of specific and general information in the context of memory schemas for events or ‘scripts.’ Its findings will be of general interest to neuroscientists and cognitive psychologists who work both with typical young adults (as studied in this paper) and in particular populations (e.g., development and/or aging; patients with brain damage). The work is particularly comprehensive in how it links both specific and general narrative representation at both encoding and retrieval with later memory behavior. This comprehensive treatment is a notable strength.

---

## [Decision Letter]

**Decision letter after peer review:**

Thank you for submitting your article "Schema representations in distinct brain networks support narrative memory during encoding and retrieval" for consideration by *eLife*. Your article has been reviewed by 2 peer reviewers, and the evaluation has been overseen by a David Badre as the Reviewing Editor, and Chris Baker as the Senior Editor. The following individual involved in review of your submission has agreed to reveal their identity: Brice Kuhl (Reviewer #1).

Essential revisions:

The reviewers were positive overall about the manuscript, and though they identified some points of weakness, they felt these could be addressed in a revision. Their essential comments are provided here.

1. Both reviewers raised interpretational issues related to the behavioral measures of story-specific recall. I provide their comments on this point together here.

Reviewer #1 – There should be a little more consideration of why schema representations predict recall of story-specific information. Is the neural measure 'missing' the information that is actually being recalled and instead is just capturing the scaffold upon which the event-specific information is laid? This point is acknowledged in the Discussion, but it feels like it warrants a bit more emphasis (i.e., not to just note that this is a limitation but to speculate a bit more about why schema representations are related to story-specific recall).

Reviewer #2 – From a theoretical perspective, I am struggling with the behavioral outcome measures being exclusively at the "specific" story level, and whether/how that should impact our interpretation of the findings. In other words, the behavioral outcome of interest has to do with participants' ability to recall story-specific details, and a score was given to each subject for each story to summarize the quality of their memory for that particular narrative. By necessity, of course, this means knowledge at the "schematic" level is not tested or operationalized in any way. (In fact, it would I believe be impossible to do this on a narrative-by-narrative basis.) The authors address this in their setup, discussing how a schema can be used to guide the retrieval of details, and also touch upon this in the Discussion (lines 404-410). However, I am struggling with the contrast between the memory ~ encoding and memory ~ reinstatement findings being whopping and widespread for the story neural representation (Figure 3A, C), and much smaller (and nonsignificant in many ROIs) for the schema neural representation (Figure 3B, D). Is this showing us that (detailed, specific) story representation supports recall of (detailed, specific) memories, and (general, abstracted) schema representation does not? Does that mean schema representation does not relate to memory, or just that it doesn't relate to *specific* memory (i.e., but could have in theory been related to schematic memory, had that been tested)? I suppose from some vantage points, it could be viewed as merely a replication of many other findings that representing specific memories at either encoding or retrieval is helpful for recall of those details. And similarly, one could argue that schema representations haven't been given a fair shake because the behavior was tested at a different level of specificity. In other words, in their analysis for Figure 3 B and D the authors separately considered the relationship between schema representation and behavior, without simultaneously considering the level of specific story representation, which is a bit hard to reconcile with the framework that schemas would guide retrieval via reinstatement of specific details (i.e., theoretically, should we expect that they can support detail recall on their own? or should it be that schema representation supports specific memory, but only when detail recall is also high?). With the exception of the mediation analysis in Figure 5 (which I think does speak to this point in a nice way), the earlier, primary analyses do not take this complexity into account. To be clear, I am not sure answering these questions requires new analyses, and am not asking the authors to change their approach. I am more hoping the authors could provide us more of their thoughts on these points in the paper and perhaps soften their conclusions if appropriate.

2. The regions that predicted behavior during encoding are described as "surprisingly distinct" from those that predicted behavior during retrieval. However, direct statistical comparisons of encoding vs. retrieval are not reported. Rather, it is more of an informal, qualitative comparison of which regions turned up in each analysis. While the results are appropriately described (statements are carefully worded), I do think the conclusions that can be drawn about differences between encoding and retrieval are somewhat limited without more direct statistical comparisons. For example, differences between anterior and posterior hippocampus (in relation to how schema vs. story-specific representations predict behavior) are supported by a direct statistical contrast of these regions.

3. The clustering analysis that is depicted in Figure 5 is a bit convoluted or hard to follow. For example, I found it very difficult to map the figure legend (particularly part A) to the actual figure. More generally, the rationale for using the clustering analysis could benefit from some elaboration.

4. The rationale (including potential advantages and disadvantages) for relying on intersubject analyses should be elaborated. In particular, it seems very possible-or very likely-that intersubject consistency is going to differ during encoding (when the pacing and order of information is dictated by the stimuli) compared to recall (when the pacing, details, and order could be idiosyncratic). This just feels like a non-trivial aspect of the design, yet it receives virtually no consideration.

5. It was not clear how the audio vs. video difference was worked into the analysis, or why for the schema scores, different-modality patterns were not also considered. It would seem as though comparing patterns derived from the presentation of video vs. audio as part of the schema measure would allow the researchers to get around potential confounds like visual presentation of the same type of stimuli across narratives of the same type to drive the "schema" representation (e.g., restaurant videos presumably show a lot of the same types of objects as one another, but those same objects would not be presented visually in the audio clips). Similarly, perhaps audio clips contained similar words for a given schema. It seems as though airport 1 video being more similar to airport 1-4 audio than it is to restaurant 1-4 audio (all different modality comparisons) would be a powerful way to demonstrate schema representation (I believe the authors have done this in past work; Baldassano et al. 2018 J Neuro). In any case, I think this detail and reasoning should be added to the main paper, and potentially worked into the visualizations.

6. It was unclear from the methods how the models relating neural scores with behavioral performance were set up. It sounds as though perhaps the researchers ran a simple linear regression, such that all participants' data was combined into a single model but subjects were not treated as random effects. If this were the case, then variability in memory performance across subjects is going to contribute to the estimate of the within-subject relationship between neural scores and memory performance on a story-by-story basis. It seems from the paper as though the authors are more interested in the within-subject variability. Can the authors clarify this point (e.g., by expanding the methods section beginning on line 585)?

7. Some references to figure panels that don't appear to exist. For example, there is reference to figure 4A and 4B, but figure 4 does not have any sub-panels.

8. Line 52 – the theoretical importance of free recall was a little unclear from the writing.

9. Figure 1 panel – just a suggestion that it may be helpful to orient correlation matrix schematic on the right hand side in the same was as the one in panel C, because I believe the "encoding templates" (currently on the y-axis for panel D) would be the same as the "n-1 encoding story" on the panel C schematic. It may help comprehension to put these both on the x-axis (for example) and/or match the labels and/or draw attention to this parallel in the legend. Could "Recall story" also be described as "subj i Recall story" to match the earlier panel?

10. It might be worth explicitly noting earlier in the paper (specifically, before the reader gets to Figure 1) that the videos are in fact taken from real feature films, rather than constructed narratives. This was a little ambiguous from the phrasing in the Introduction.

---

## [Author Response]

Essential revisions:The reviewers were positive overall about the manuscript, and though they identified some points of weakness, they felt these could be addressed in a revision. Their essential comments are provided here.1. Both reviewers raised interpretational issues related to the behavioral measures of story-specific recall. I provide their comments on this point together here.Reviewer #1 – There should be a little more consideration of why schema representations predict recall of story-specific information. Is the neural measure 'missing' the information that is actually being recalled and instead is just capturing the scaffold upon which the event-specific information is laid? This point is acknowledged in the Discussion, but it feels like it warrants a bit more emphasis (i.e., not to just note that this is a limitation but to speculate a bit more about why schema representations are related to story-specific recall).Reviewer #2 – From a theoretical perspective, I am struggling with the behavioral outcome measures being exclusively at the "specific" story level, and whether/how that should impact our interpretation of the findings. In other words, the behavioral outcome of interest has to do with participants' ability to recall story-specific details, and a score was given to each subject for each story to summarize the quality of their memory for that particular narrative. By necessity, of course, this means knowledge at the "schematic" level is not tested or operationalized in any way. (In fact, it would I believe be impossible to do this on a narrative-by-narrative basis.) The authors address this in their setup, discussing how a schema can be used to guide the retrieval of details, and also touch upon this in the Discussion (lines 404-410). However, I am struggling with the contrast between the memory ~ encoding and memory ~ reinstatement findings being whopping and widespread for the story neural representation (Figure 3A, C), and much smaller (and nonsignificant in many ROIs) for the schema neural representation (Figure 3B, D). Is this showing us that (detailed, specific) story representation supports recall of (detailed, specific) memories, and (general, abstracted) schema representation does not? Does that mean schema representation does not relate to memory, or just that it doesn't relate to specific memory (i.e., but could have in theory been related to schematic memory, had that been tested)? I suppose from some vantage points, it could be viewed as merely a replication of many other findings that representing specific memories at either encoding or retrieval is helpful for recall of those details. And similarly, one could argue that schema representations haven't been given a fair shake because the behavior was tested at a different level of specificity. In other words, in their analysis for Figure 3 B and D the authors separately considered the relationship between schema representation and behavior, without simultaneously considering the level of specific story representation, which is a bit hard to reconcile with the framework that schemas would guide retrieval via reinstatement of specific details (i.e., theoretically, should we expect that they can support detail recall on their own? or should it be that schema representation supports specific memory, but only when detail recall is also high?). With the exception of the mediation analysis in Figure 5 (which I think does speak to this point in a nice way), the earlier, primary analyses do not take this complexity into account. To be clear, I am not sure answering these questions requires new analyses, and am not asking the authors to change their approach. I am more hoping the authors could provide us more of their thoughts on these points in the paper and perhaps soften their conclusions if appropriate.

We thank the reviewers for these comments. Both R1 and R2 raise important questions about how to interpret the relationship between neural schema scores and recall behavior (which is relatively localized in the brain) in light of the more widespread relationship between neural story scores and recall behavior – do these (more localized) neural schema effects reflect the same underlying process as the neural story effects, or is neural schema representation contributing something “extra” to memory performance, beyond what is provided by neural story effects? Inspired by R2’s comment about analyzing story and schema effects together, we ran a new analysis to address this issue. Specifically, we used a nested model comparison analysis (reported in Figure 3 – Supplementary Figure 2) to assess whether neural schema scores improve our ability to predict behavioral memory for specific details, beyond how well we can predict memory based on neural story scores alone. This analysis revealed that the neural schema scores do indeed provide this “extra” benefit in predicting recall, beyond what we can do using neural story scores alone. We have added this new analysis to the paper, along with discussion of possible mechanisms that could give rise to this finding (in which we both speculate about possible mechanisms and also acknowledge limitations of our analysis).

In accordance with this change, we added a new subsection to the Methods titled “Benefit of schema representation for detail memory”:

“In order to determine the degree to which neural schema scores at encoding or retrieval improved prediction of behavioral memory for story-specific information (beyond what could be predicted from neural story scores alone), we compared a single-regressor baseline model with story scores as the sole predictor against a nested model with schema scores as an additional predictor (Figure 3 – Supp 2). […] Extracted p-values were converted to q-values using AFNI's FDR-correction.”

Finally, we have amended the Discussion section in a few places to discuss these results and possible mechanisms that could give rise to these results:

“Importantly, follow-up analyses revealed that, both at encoding and at recall, our ability to predict recall behavior based on neural schema scores and neural story scores (together) was better than prediction based on neural story scores alone (Figure 3 – Supp 2). […] The results of the present study do not speak to the validity of either theory – more specialized designs will be needed to test these accounts.”

We also modified a later paragraph of the discussion:

“… our results extend the literature by revealing a within-subjects (across-story) relationship between the activation of anterior mPFC schema representations at encoding and memory for details of temporally-extended naturalistic stimuli. These findings are compatible with the idea that mPFC might provide a schematic scaffold to which detailed representations are attached at encoding (Gilboa and Marlatte 2017; Schlichting and Preston 2015; van Kesteren et al., 2012).”

2. The regions that predicted behavior during encoding are described as "surprisingly distinct" from those that predicted behavior during retrieval. However, direct statistical comparisons of encoding vs. retrieval are not reported. Rather, it is more of an informal, qualitative comparison of which regions turned up in each analysis. While the results are appropriately described (statements are carefully worded), I do think the conclusions that can be drawn about differences between encoding and retrieval are somewhat limited without more direct statistical comparisons. For example, differences between anterior and posterior hippocampus (in relation to how schema vs. story-specific representations predict behavior) are supported by a direct statistical contrast of these regions.

Thank you for noting this opportunity to strengthen the claims of the paper by quantitatively comparing the effects at encoding vs. retrieval. We have added a new analysis to compare the schematic model coefficients at encoding vs those at retrieval in a searchlight analysis across cortex. We have added a new paragraph starting on line 246 of the Results section to introduce the analysis:

“To formally compare how schematic representations at encoding and retrieval predict subsequent memory differently, we ran a model coefficient comparison analysis in which we contrasted the encoding coefficients against the reinstatement coefficients (Figure 4B). This analysis corroborated our conjunction analysis (Figure 4A), revealing one set of regions that had significantly stronger relationships to subsequent memory at encoding than at recall (portions of left visual cortex, left posterior temporal sulcus, right prostriata, entorhinal cortex, left subcentral cortex, postcentral sulcus, frontal gyrus, and portions of anterior mPFC) and another set of regions that had significantly stronger relationships to subsequent memory at recall than encoding (visual cortex, SPL, clusters in posterior parietal regions, SFG, regions in frontal gyrus, and PHC).”

These results have been added to panel B on Figure 4.

We have also modified the Methods section “Schema representation and subsequent behavior” by adding a new paragraph starting on line 725 with the following:

“To quantify how schema representations influenced memory at both encoding and recall (Figure 4B), we compared their model coefficients (encoding – reinstatement). Statistical significance was computed via 2-sided non-parametric testing of a null distribution of differences between model coefficients at encoding vs reinstatement. P-values were then converted to q-values with AFNI's FDR-correction.”

3. The clustering analysis that is depicted in Figure 5 is a bit convoluted or hard to follow. For example, I found it very difficult to map the figure legend (particularly part A) to the actual figure. More generally, the rationale for using the clustering analysis could benefit from some elaboration.

Thank you for bringing this confusion up. The source of confusion may have been due in part to a labeling error in the caption, which we have corrected. To further clarify the caption, we changed some of the wording and added location pointers in the figure descriptions. See changes below:

“We performed post-hoc k-means (k=2) clustering within a bilateral mPFC ROI to identify subregions showing different profiles of results across our 8 searchlight analyses; that is, each searchlight was assigned an 8-dimensional vector reflecting the scores assigned to that region in the 8 searchlight analyses, and clustering was done on these 8-dimensional vectors (Figure 2 and Figure 3). […] Consistent with our searchlight results, we find that encoding schema scores in voxels in the anterior cluster (c0) were associated with increased memory performance (top-right graph).”

We also added text to the Results section expanding on our motivation for the clustering analysis starting right after the first sentence of the “K-means clustering” section.

“Given that our primary searchlight analysis revealed a relationship between encoding schema scores and subsequent memory in the most anterior part of mPFC, but this relationship was not significant in our whole-mPFC ROI (Figure 3B), we decided to perform a post-hoc clustering analysis to determine whether subregions of mPFC contributed to memory differently.”

4. The rationale (including potential advantages and disadvantages) for relying on intersubject analyses should be elaborated. In particular, it seems very possible-or very likely-that intersubject consistency is going to differ during encoding (when the pacing and order of information is dictated by the stimuli) compared to recall (when the pacing, details, and order could be idiosyncratic). This just feels like a non-trivial aspect of the design, yet it receives virtually no consideration.

Thank you for highlighting this point. We have added supplementary figures to both Figure 2 and Figure 3 where we ran the same analyses under a within-subjects approach. We modified multiple sections of the Methods to account for this inclusion.

In the Methods subsection “Encoding similarity matrix” we added a second paragraph that introduces the within-subjects approach for the encoding similarity matrix:

“Because prior work has shown that between-participant analyses can uncover shared neural event representations across participants (Baldassano, Hasson, and Norman, 2018; Baldassano et al., 2017; Chen et al., 2017; Zadbood, Chen, Leong, Norman, and Hasson, 2017) and boost signal-to-noise-ratio (Simony et al., 2016), we used an across-subject encoding similarity matrix as our default analysis. […] This results in a 16x16 symmetric correlation matrix with valid off-diagonal entries, corresponding to the event-specific similarity of one story to another, and ones in the principal diagonal, corresponding to the perfect similarity of a story's patterns with itself.”

Similarly, we added another paragraph in the “Reinstatement similarity matrix” subsection:

“Just as we created a within-subjects version of our encoding similarity matrix, we also created a within-subjects version of our reinstatement similarity matrix. […] This yielded a 16x16 (within-subject) encoding-recall similarity matrix per participant.”

To clarify that a story-score cannot be computed at encoding when using a within-subjects approach, we added a paragraph in the Methods where story scores are explained:

“Note that, for the within-subjects *encoding* similarity matrix described earlier, a story score cannot be computed because the similarity of a participant's encoding template to itself is always going to be 1 (Figure 2 – Supp 1). Therefore, for the within-subjects encoding similarity matrix, only a schema score was calculated, but for the within-subjects reinstatement similarity matrix, both story and schema scores were computed.”

Similarly, we added a sentence at the end of the “Predicting behavioral performance from neural scores” section of the Methods:

“These same analyses were re-run using story and schema scores derived using a within-subjects approach, and also using the across-modality contrast described previously.”

Figure 2 – Supplementary Figure 1 includes the searchlight and specific ROI results of the within-subjects approach. New figure caption below:

“Neural story and schema strength during encoding and retrieval in whole-brain and specific cortical and hippocampal ROIs, computed within-subjects. […] Starred points indicate significant differences after Bonferroni correction for multiple comparisons.”

Figure 3 – Supplementary Figure 3 includes the searchlight and specific-ROI results for the analyses in which within-subjects encoding and reinstatement scores were used to predict memory. New figure caption below:

“Predicting behavioral memory for story details with neural measures from encoding and recall within-subjects. […] Point colors indicate directionality of prediction with red and blue for positive and negative associations, respectively.”

Finally, brief discussion of these results can be found in the new Discussion section “Within-subject and across-modality supplementary results”:

“Prior work has also shown that between-subject analyses can improve signal-to-noise ratios while revealing shared representations across people (Baldassano, Hasson, and Norman, 2018; Baldassano et al., 2017; Chen et al., 2017; Zadbood, Chen, Leong, Norman, and Hasson, 2017; Simony et al., 2016). […] Also, the negative relationship between encoding schema scores in posterior hippocampus and subsequent memory was no longer significant (but the positive relationship between reinstatement schema scores in posterior hippocampus and subsequent memory was still significant).”

5. It was not clear how the audio vs. video difference was worked into the analysis, or why for the schema scores, different-modality patterns were not also considered. It would seem as though comparing patterns derived from the presentation of video vs. audio as part of the schema measure would allow the researchers to get around potential confounds like visual presentation of the same type of stimuli across narratives of the same type to drive the "schema" representation (e.g., restaurant videos presumably show a lot of the same types of objects as one another, but those same objects would not be presented visually in the audio clips). Similarly, perhaps audio clips contained similar words for a given schema. It seems as though airport 1 video being more similar to airport 1-4 audio than it is to restaurant 1-4 audio (all different modality comparisons) would be a powerful way to demonstrate schema representation (I believe the authors have done this in past work; Baldassano et al. 2018 J Neuro). In any case, I think this detail and reasoning should be added to the main paper, and potentially worked into the visualizations.

This is a great observation. As suggested, we have added an extra supplementary figure to both Figure 2 and Figure 3 in which we re-run the same schema analyses but across modalities.

We also added another paragraph in this section in which we explain how we computed a schema score across-modality:

“For completeness, we also computed an across-modality schema score for each story, by contrasting the average of that story's similarity to other stories from the same schema and different modality with the average of that story's similarity to other stories from the other schema and different modality. […] Statistical significance was determined as described in the previous paragraph.”

To clarify how these scores were used when predicting behavior, a sentence at the end of the “Predicting behavioral performance from neural scores” section of the Methods was added:

“These same analyses were re-run using story and schema scores derived using a

within-subjects approach, and also using the across-modality schema scores described previously.”

Figure 2 – Supplementary Figure 2 includes the searchlight and specific ROI results of the across-modality approach:

“Neural schema strength during encoding and retrieval in whole-brain and specific cortical and hippocampal ROIs, computed across-modality. […] Starred points indicate significant differences after Bonferroni correction for multiple comparisons.”

Figure 3 – Supplementary Figure 4 includes the searchlight and specific ROI results of the across-modality approach:

“Predicting behavioral memory for story details with across-modality neural schema measures from encoding and recall. […] Point colors indicate directionality of prediction with red and blue for positive and negative associations, respectively.”

Finally, brief discussion of these results can be found in the new Discussion section “Within-subject and across-modality supplementary results”:

“Baldassano et al., (2018) found that schema representation at encoding across many of the same regions reported here (Figure 2) was modality-independent. […] When we related these across-modality neural schema scores to behavior (Figure 3 Supp4), we again found positive relationships to memory performance in mPFC at encoding; among a sparser set of regions at recall, we also found a positive relationship to memory in anterior hippocampus, suggesting that hippocampus can generalize across both *content* (i.e., shared structure across stories) and the surface form of content (i.e., modality).”

6. It was unclear from the methods how the models relating neural scores with behavioral performance were set up. It sounds as though perhaps the researchers ran a simple linear regression, such that all participants' data was combined into a single model but subjects were not treated as random effects. If this were the case, then variability in memory performance across subjects is going to contribute to the estimate of the within-subject relationship between neural scores and memory performance on a story-by-story basis. It seems from the paper as though the authors are more interested in the within-subject variability. Can the authors clarify this point (e.g., by expanding the methods section beginning on line 585)?

Thank you for raising this point. To clarify our approach, we fit a linear regression to all but one subject, and then use this model to predict story-by-story behavioral scores in the last subject (repeated for all choices of the held-out subject). We computed statistical significance by creating a null distribution in which story scores were shuffled within each participant before running the analysis. Therefore any significant results we observe must be due to within-subject (across stories) relationships between neural measures and behavior. Using a mixed-effects model could potentially increase the sensitivity of our regression analysis, by removing across-subject variance when measuring the relationship between neural and behavioral measures. However, since our analysis requires fitting this model for each searchlight, permutation, and left-out subject, it was computationally infeasible to fit this kind of model. Note that our choice to use a simpler model should lead only to false negatives (reducing sensitivity to subtle effects) and not to false positives in our results.

We have added an explanation before the last sentence of the second paragraph in the subsection starting at line 690 labeled “Predicting behavioral performance from neural scores” to clarify this point:

“Given this type of permutation, any significant results observed are due to within-subject (but across-stories) relationships between neural measures and behavior. […] Note that our choice to use a simpler model should lead only to false negatives (reducing sensitivity to subtle effects) and not to false positives in our results.”

7. Some references to figure panels that don't appear to exist. For example, there is reference to figure 4A and 4B, but figure 4 does not have any sub-panels.

Thank you so much for pointing this out! We have removed the incorrect placement of these references and have double checked all caption references to avoid any future confusion.

8. Line 52 – the theoretical importance of free recall was a little unclear from the writing.

In order to clarify the difference between traditional memory recall tasks that are not self-paced and tend to assist participants in their memory search with rich associative cues or images, we have amended line 53 to include this distinction.

“Lastly, because existing paradigms have mostly tested memory with recognition or short associative recall tasks, the neural mechanisms of how schemas are instantiated and maintained during unconstrained memory search for naturalistic events have not been thoroughly explored; our use of free recall allowed us to address this gap in the literature.”

9. Figure 1 panel – just a suggestion that it may be helpful to orient correlation matrix schematic on the right hand side in the same was as the one in panel C, because I believe the "encoding templates" (currently on the y-axis for panel D) would be the same as the "n-1 encoding story" on the panel C schematic. It may help comprehension to put these both on the x-axis (for example) and/or match the labels and/or draw attention to this parallel in the legend. Could "Recall story" also be described as "subj i Recall story" to match the earlier panel?

This is a great suggestion and we have switched the axes as well as re-labeled them to be “N Encoding Templates (Stories)” for the x-axis label and “subj i Recall Story” for the y-axis label.

10. It might be worth explicitly noting earlier in the paper (specifically, before the reader gets to Figure 1) that the videos are in fact taken from real feature films, rather than constructed narratives. This was a little ambiguous from the phrasing in the Introduction.

To address this ambiguity, we amended the introduction clarifying that the movies and audio narratives are taken from real feature films.

“The present study builds on our prior work (Baldassano
et al., 2018), in which participants were scanned as they watched movies or listened to audio-adaptations of pre-existing films, half of which followed a restaurant script and half of which followed an airport script.”